# Minimax Regret for Stochastic Shortest Path

**Alon Cohen**
Tel-Aviv University and Google Research, Tel Aviv
aloncohen@google.com

**Yonathan Efroni**
Microsoft Research, New York
jonathan.efroni@gmail.com

**Yishay Mansour**
Tel-Aviv University and Google Research, Tel Aviv
mansour@tau.ac.il

**Aviv Rosenberg**
Tel-Aviv University
avivros007@gmail.com

## Abstract

We study the Stochastic Shortest Path (SSP) problem in which an agent has to reach a goal state in minimum total expected cost. In the learning formulation of the problem, the agent has no prior knowledge about the costs and dynamics of the model. She repeatedly interacts with the model for $K$ episodes, and has to minimize her regret. In this work we show that the minimax regret for this setting is $\widetilde{O}(\sqrt{(B_\star^2 + B_\star)|S||A|K})$ where $B_\star$ is a bound on the expected cost of the optimal policy from any state, $S$ is the state space, and $A$ is the action space. This matches the $\Omega(\sqrt{B_\star^2|S||A|K})$ lower bound of Rosenberg et al. [2020] for $B_\star \geq 1$, and improves their regret bound by a factor of $\sqrt{|S|}$. For $B_\star < 1$ we prove a matching lower bound of $\Omega(\sqrt{B_\star|S||A|K})$. Our algorithm is based on a novel reduction from SSP to finite-horizon MDPs. To that end, we provide an algorithm for the finite-horizon setting whose leading term in the regret depends polynomially on the expected cost of the optimal policy and only logarithmically on the horizon.

## 1  Introduction

We study the stochastic shortest path (SSP) problem in which an agent aims to reach a predefined goal state while minimizing her total expected cost. This is one of the most basic models of reinforcement learning (RL) that includes both finite-horizon and discounted Markov Decision Processes (MDPs) as special cases. In addition, SSP captures a wide variety of realistic scenarios such as car navigation, game playing and drone flying.

We study an online version of SSP in which both the immediate costs and transition distributions of the model are initially unknown to the agent. The agent interacts with the model for $K$ episodes, in each of which she attempts to reach the goal state with minimal cumulative cost. A main challenge in the online model is found when instantaneous costs are small. For example, any learning algorithm that attempts to myopically minimize the accumulated costs might get caught in a cycle with zero cost and never reach the goal state. Nonetheless, even if the costs are not zero, only very small, the agent must be able to trade off the need to minimize costs with that of reaching the goal quickly.

The online setting was originally suggested by Tarbouriech et al. [2020] who gave an algorithm with $\widetilde{O}(K^{2/3})$ regret guarantee. In a follow-up work, Rosenberg et al. [2020] improved the previous bound to $\widetilde{O}(B_\star|S|\sqrt{|A|K})$, where $S$ is the state space, $A$ is the action space, and $B_\star$ is an upper bound on the total expected cost of the optimal policy when initialized at any state. Rosenberg et al. [2020] also provide a lower bound of $\Omega(B_\star\sqrt{|S||A|K})$ – leaving a gap of $\sqrt{|S|}$ between the upper and lower bounds. In this work, unlike the previously mentioned works that assume the cost function is deterministic

35th Conference on Neural Information Processing Systems (NeurIPS 2021).

and known, we consider the case where the costs are i.i.d. and initially unknown. We prove upper and lower bounds for this case, proving that the optimal regret is of order $\widetilde{\Theta}(\sqrt{(B_\star^2 + B_\star)|S||A|K})$.

The algorithms of both Tarbouriech et al. [2020], Rosenberg et al. [2020] were based on a direct application of the "Optimism in the Face of Uncertainty" principle to the SSP model, following the ideas behind the UCRL2 algorithm [Jaksch et al., 2010] for average-reward MDPs. In this work we take a different approach. We propose a novel black-box reduction to finite-horizon MDPs, showing that the SSP problem is not harder than the finite-horizon setting assuming prior knowledge on the expected time it takes for the optimal policy to reach the goal state. While the reduction itself is simple, the analysis is highly nontrivial as one has to show that the goal state is indeed reached in every episode without incurring excessive costs in the process.

The idea of reducing SSP to finite-horizon was previously used by Chen et al. [2020], Chen and Luo [2021] for SSP with adversarially changing costs. However, they run one finite-horizon episode in every SSP episode and then simply try to reach the goal as fast as possible, while we restart a new finite-horizon episode every $H$ steps. This modification is what enables us to obtain the optimal and improved dependence in the number of states.

In addition, we provide a new algorithm for regret minimization in finite-horizon MDPs called ULCVI. We show that (for large enough number of episodes) its regret depends polynomially on the expected cost of the optimal policy $B_\star$, and only logarithmically on the horizon length $H$. This implies that the correct measure for the regret is the expected cost of the optimal policy and not the length of the horizon. We note that regret with logarithmic dependence in the horizon $H$ was also obtained by Zhang et al. [2020], yet they make a much stronger assumption: that the cumulative cost of *every* trajectory is bounded by 1. In contrast, we only assume that the *expected* cost of the optimal policy is bounded by some constant $B_\star$, while other policies may suffer a cost of $H$.

Our reduction, when combined with our finite-horizon algorithm ULCVI, guarantees SSP regret of $\widetilde{O}(\sqrt{(B_\star^2 + B_\star)|S||A|K})$. This matches the lower bound of Rosenberg et al. [2020] for $B_\star \geq 1$ up to logarithmic factors. However, their lower bound does not hold for $B_\star < 1$ suggesting that this is not the correct rate in this case. Indeed, we prove a tighter lower bound of $\Omega(\sqrt{B_\star|S||A|K})$ for $B_\star < 1$, showing that our regret guarantees are minimax optimal in all cases.

As a final remark we note that, following our work, Tarbouriech et al. [2021] were able to obtain a comparable regret bound for SSP without prior knowledge of the optimal policy's expected time to reach the goal state.

## 1.1 Additional related work

**Planning for stochastic shortest path.** Early work by Bertsekas and Tsitsiklis [1991] studied planning in SSPs, i.e., computing the optimal strategy efficiently when parameters are known. Under certain assumptions, they established that the optimal strategy is a deterministic stationary policy and can be computed efficiently using standard planning algorithms, e.g., Value Iteration and LP.

**Adversarial stochastic shortest path.** Rosenberg and Mansour [2020] presented stochastic shortest path with adversarially changing costs. Their regret bounds were improved by Chen et al. [2020], Chen and Luo [2021] using a reduction to online loop-free SSP (see next paragraph). As mentioned before, our reduction is different and therefore able to remove the extra $\sqrt{|S|}$ factor in the regret.

**Regret minimization in MDPs.** There is a vast literature on regret minimization in RL that mostly builds on the optimism principle. Most literature focuses on the tabular setting [Jaksch et al., 2010, Azar et al., 2017, Jin et al., 2018, Fruit et al., 2018, Zanette and Brunskill, 2019, Efroni et al., 2019, Simchowitz and Jamieson, 2019], but recently it was extended to function approximation under various assumptions [Yang and Wang, 2019, Jin et al., 2020b, Zanette et al., 2020a,b].

**Online loop-free SSP.** A different line of work considers finite-horizon MDPs with adversarially changing costs [Neu et al., 2010, 2012, Zimin and Neu, 2013, Rosenberg and Mansour, 2019b,a, Jin et al., 2020a, Cai et al., 2020, Shani et al., 2020, Lancewicki et al., 2020, Lee et al., 2020, Jin and Luo, 2020]. They refer to finite-horizon adversarial MDPs as online loop-free SSP. This is not to be confused with our setting in which the interaction between the agent and the environment ends only when (and if) the goal state is reached, and not after a fixed number of steps $H$. See Rosenberg and Mansour [2020], Chen et al. [2020] for a discussion on the differences between the models.

## 2 Preliminaries and main results

An instance of the SSP problem is defined by an MDP $\mathcal{M} = (S, A, P, c, s_{\text{init}}, g)$ where $S$ is a finite state space and $A$ is a finite action space. The agent begins at an initial state $s_{\text{init}} \in S$, and ends her interaction with $\mathcal{M}$ by arriving at the goal state $g$ (where $g \notin S$). Whenever she plays action $a$ in state $s$, she pays a cost $C \in [0, 1]$ drawn i.i.d. from a distribution with expectation $c(s, a) \in [0, 1]$ and the next state $s' \in S \cup \{g\}$ is chosen with probability $P(s' \mid s, a)$. Note that the transition function $P$ satisfies $\sum_{s' \in S \cup \{g\}} P(s' \mid s, a) = 1$ for every $(s, a) \in S \times A$.

**Proper policies.** A stationary and deterministic policy $\pi : S \mapsto A$ is a mapping that selects action $\pi(s)$ whenever the agent is at state $s$. A policy $\pi$ is called *proper* if playing according to $\pi$ ensures that the goal state is reached with probability 1 when starting from any state (otherwise it is *improper*). In SSP, the agent has two goals: (a) reach the goal state; (b) minimize the total expected cost. To facilitate the first goal, we make the basic assumption that there exists at least one proper policy. In particular, the goal state is reachable from every state, which is clearly a necessary assumption.

Any policy $\pi$ induces a *cost-to-go function* $J^\pi : S \mapsto [0, \infty]$. The cost-to-go at state $s$ is defined by $J^\pi(s) = \lim_{T \to \infty} \mathbb{E}_\pi \left[ \sum_{t=1}^{T} c(s_t, a_t) \mid s_{\text{init}} = s \right]$, where the expectation is taken w.r.t the random sequence of states generated by playing according to $\pi$ when the initial state is $s$. For a proper policy $\pi$, it follows that $J^\pi(s)$ is finite for all $s \in S$. However, note that $J^\pi(s)$ may be finite even if $\pi$ is improper. We additionally denote by $T^\pi(s)$ the expected time it takes for $\pi$ to reach $g$ starting at state $s$; in particular, if $\pi$ is proper then $T^\pi(s)$ is finite for all $s \in S$, and if $\pi$ is improper there must exist some state $s$ such that $T^\pi(s) = \infty$.

**Learning formulation.** Here, the agent does not have any prior knowledge of the cost function $c$ or transition function $P$. She interacts with the model in episodes: each episode starts at the fixed initial state $s_{\text{init}}$,[1] and ends when the agent reaches the goal state $g$ (note that she might *never* reach the goal state). Success is measured by the agent's regret over $K$ such episodes, that is the difference between her total cost over the $K$ episodes and the total expected cost of the optimal proper policy:

$$R_K = \sum_{k=1}^{K} \sum_{i=1}^{I^k} C_i^k - K \cdot \min_{\pi \in \Pi_{\text{proper}}} J^\pi(s_{\text{init}}),$$

where $I^k$ is the time it takes the agent to complete episode $k$ (which may be infinite), $C_i^k$ is the cost suffered in the $i$-th step of episode $k$ when the agent visited state-action pair $(s_i^k, a_i^k)$, and $\Pi_{\text{proper}}$ is the set of all stationary, deterministic and proper policies (that is not empty by assumption). In the case that $I^k$ is infinite for some $k$, we define $R_K = \infty$.

We denote the optimal proper policy by $\pi^\star$, $J^{\pi^\star}(s_{\text{init}}) = \arg\min_{\pi \in \Pi_{\text{proper}}} J^\pi(s_{\text{init}})$. Moreover, let $B_\star > 0$ be an upper bound on the values of $J^{\pi^\star}$ and let $T_\star > 0$ be an upper bound on the times $T^{\pi^\star}$, i.e., $B_\star \geq \max_{s \in S} J^{\pi^\star}(s)$ and $T_\star \geq \max_{s \in S} T^{\pi^\star}(s)$. Finally, let $D = \max_{s \in S} \min_{\pi \in \Pi_{\text{proper}}} T^\pi(s)$ be the SSP-diameter, and note that $B_\star \leq D \leq T_\star$.

### 2.1 Summary of our results

In Section 3 we present a novel black-box reduction from SSP to finite-horizon MDPs (Algorithm 1), that yields $\sqrt{K}$ regret bounds when combined with a certain class of optimistic algorithms for regret minimization in finite-horizon MDPs that we call *admissible* (Definition 1). The regret analysis for the reduction is described in Section 4, and in Section 5 we present an admissible algorithm for regret minimization in finite-horizon MDPs called ULCVI. We show that it guarantees the following optimal regret in the finite-horizon setting (stated formally in Theorem 5.1). Note that (for large enough number of episodes) this bound depends only on the expected cost of the optimal policy and not on the horizon $H$.

**Theorem 2.1.** *Running* ULCVI *(Algorithm 2 in Section 5) in a finite-horizon MDP guarantees, with probability at least* $1 - \delta$, *a regret bound of*

$$O\left( \sqrt{(B_\star^2 + B_\star) |S||A|M} \log \frac{MH|S||A|}{\delta} + H^4 B_\star^{-1} |S|^2 |A| \log^{3/2} \frac{MH|S||A|}{\delta} \right),$$

---

[1]The initial state is fixed for simplicity of presentation, but it can be chosen adversarially at the beginning of every episode. Without any change to the algorithm or analysis, the same guarantees hold.

*for any number of episodes $M \geq 1$ simultaneously.*

Combining `ULCVI` with our reduction yields the following minimax optimal regret bound for SSP.

**Theorem 2.2.** *Running the reduction in Algorithm 1 with the finite-horizon regret minimization algorithm `ULCVI` ensures, with probability at least $1 - \delta$,*

$$R_K = O\left( \sqrt{(B_\star^2 + B_\star)|S||A|K} \log \frac{KT_\star|S||A|}{\delta} + T_\star^5 B_\star^{-2}|S|^2|A| \log^6 \frac{KT_\star|S||A|}{\delta} \right).$$

**Remark 1.** An important observation is that this regret bound is meaningful even for small $K$. Unlike finite-horizon MDPs, where linear regret is trivial, in SSP ensuring finite regret is not easy. Our regret bound also implies that if we play for only one episode, i.e., we are only interested in the time it takes to reach the goal state, then it will take us at most $\widetilde{O}(T_\star^5 B_\star^{-2}|S|^2|A|)$ time steps to do so.

**Remark 2.** Note that our algorithm needs to know an upper bound on $T_\star$ in advance. However, if all costs are strictly positive (i.e., at least $c_{\min} > 0$), then there is a trivial upper bound of $B_\star/c_{\min}$. In this case, our algorithm keeps an optimal regret bound for large enough $K$, since the bound on $T_\star$ only appears in the additive factor. Some previous work used a perturbation argument to generalize their results from the $c_{\min}$ case to general costs [Tarbouriech et al., 2020, Rosenberg et al., 2020, Rosenberg and Mansour, 2020]. In our case, it will not work since the dependence on $1/c_{\min}$ in the additive term is too large. This may be an inherent shortcoming of using finite-horizon reduction to solve SSPs, as it also appears in the works of Chen et al. [2020], Chen and Luo [2021] for the adversarial setting.

**Remark 3.** In practice, one can think of $T_\star$ as a parameter of the algorithm that controls computational complexity and the number of steps to complete $K$ episodes. By choosing the parameter $T_\star = x$ for example, we can guarantee that the regret bound of Theorem 2.2 holds against the best proper policy with expected time to the goal of at most $x$ (assuming there exists one), and we can also guarantee that the total computational complexity of the algorithm is $\widetilde{O}(x \log K)$ (see Remark 5). Furthermore, the algorithm will take at most $\widetilde{O}(xK + poly(x, |S|, |A|))$ steps to complete $K$ episodes.

**Remark 4.** While the additive term in our regret bound is standard for most cases, it becomes large when $B_\star$ is extremely small because of the dependence in $B_\star^{-1}$. This was not an issue in previous work [Tarbouriech et al., 2020, Rosenberg et al., 2020] since they assumed that the costs are deterministic and known. We believe that this dependence is an artifact of our analysis that may be avoided with a more careful definition of $\omega_{\mathcal{A}}$ (see Definition 1) that depends on the actual cost in each state-action pair and not just $B_\star$. Nevertheless, the main focus of this paper is on establishing that the minimax optimal regret for SSP is $\widetilde{\Theta}(\sqrt{(B_\star^2 + B_\star)|S||A|K})$, and not on optimizing lower order terms. By that we also show that this is the minimax optimal regret for finite-horizon which is independent of the horizon $H$ (up to logarithmic factors). Tightening the additive term and eliminating its dependence in $B_\star^{-1}$ is left as an interesting future direction.

In Appendix D we prove that our regret bound is indeed minimax optimal. To complement the $\Omega(B_\star\sqrt{|S||A|K})$ lower bound of Rosenberg et al. [2020] that assumes $B_\star \geq 1$, we provide the following tighter lower bound for the case that $B_\star < 1$.

**Theorem 2.3.** *Let $B_\star \leq \frac{1}{2}$. There exists an SSP problem instance $\mathcal{M} = (S, A, P, c, s_{init}, g)$ in which $J^{\pi^\star}(s) \leq B_\star$ for all $s \in S$, $|S| \geq 2$, $|A| \geq 2$, $K \geq B_\star|S||A|$, such the expected regret of any learner after $K$ episodes satisfies*

$$\mathbb{E}[R_K] \geq \frac{1}{32}\sqrt{B_\star|S||A|K}.$$

## 3 A black-box reduction from SSP to finite-horizon

Our algorithm takes as input an algorithm $\mathcal{A}$ for regret minimization in finite-horizon MDPs, and uses it to perform a black-box reduction. The algorithm is depicted below as Algorithm 1.

The algorithm breaks the individual time steps that comprise each of the $K$ episodes into *intervals* of $H$ time steps. If the agent reaches the goal state before $H$ time steps, we simply assume that she stays in $g$ until $H$ time steps are elapsed. We see each interval as one episode of a finite-horizon model $\widehat{\mathcal{M}} = (\widehat{S}, A, \widehat{P}, H, \hat{c}, \hat{c}_f)$, where $\widehat{S} = S \cup \{g\}$ and $\hat{c}_f : \widehat{S} \to \mathbb{R}$ is a set of terminal costs defined by

$\hat{c}_f(s) = 8B_\star \mathbb{I}\{s \neq g\}$, where $\mathbb{I}\{s \neq g\}$ is the indicator function that equals 1 if $s \neq g$ and 0 otherwise. Moreover, $\widehat{P}, \hat{c}$ are the natural extensions of $P, c$ to the goal state. That is, $\hat{c}(s, a) = c(s, a)\mathbb{I}\{s \neq g\}$ and

$$\widehat{P}(s' \mid s, a) = \begin{cases} P(s' \mid s, a), & s \neq g; \\ 1, & s = g, s' = g; \\ 0, & s = g, s' \neq g. \end{cases}$$

The horizon $H$ (which we will set to be roughly $T_\star$) is chosen such that the optimal SSP policy will reach the goal state in $H$ time steps with high probability (recall that the expected hitting time of the optimal policy is bounded by $T_\star$). The additional terminal cost is there to encourage the agent to reach the goal state within $H$ steps, which otherwise is not necessarily optimal with respect to the planning horizon.

---

**Algorithm 1** REDUCTION FROM SSP TO FINITE-HORIZON MDP

---

1: **input:** state sapce $S$, action space $A$, initial state $s_{\text{init}}$, goal state $g$, confidence parameter $\delta$, number of episodes $K$, bound on the expected cost of the optimal policy $B_\star$, bound on the expected time of the optimal policy $T_\star$ and algorithm $\mathcal{A}$ for regret minimization in finite-horizon MDPs.

2: **initialize** $\mathcal{A}$ with state space $\widehat{S} = S \cup \{g\}$, action space $A$, horizon $H = 8T_\star \log(8K)$, confidence parameter $\delta/4$, terminal costs $\hat{c}_f(s) = 8B_\star \mathbb{I}\{s \neq g\}$ and bound on the expected cost of the optimal policy $9B_\star$.

3: **initialize** intervals counter $m \leftarrow 0$ and time steps counter $t \leftarrow 1$.

4: **for** $k = 1, \ldots, K$ **do**

5:      set $s_t \leftarrow s_{\text{init}}$.

6:      **while** $s_t \neq g$ **do**

7:          set $m \leftarrow m + 1$, feed initial state $s_t$ to $\mathcal{A}$ and obtain policy $\pi^m = \{\pi_h^m : \widehat{S} \to A\}_{h=1}^H$.

8:          **for** $h = 1, \ldots, H$ **do**

9:              play action $a_t = \pi_h^m(s_t)$, suffer cost $C_t \sim c(s_t, a_t)$, and set $s_h^m = s_t, a_h^m = a_t, C_h^m = C_t$.

10:             observe next state $s_{t+1} \sim P(\cdot \mid s_t, a_t)$ and set $t \leftarrow t + 1$.

11:             **if** $s_t = g$ **then**

12:                 pad trajectory to be of length $H$ and BREAK.

13:             **end if**

14:          **end for**

15:          set $s_{H+1}^m = s_t$.

16:          feed trajectory $U^m = (s_1^m, a_1^m, \ldots, s_H^m, a_H^m, s_{H+1}^m)$ and costs $\{C_h^m\}_{h=1}^H$ to $\mathcal{A}$.

17:      **end while**

18: **end for**

---

The algorithm $\mathcal{A}$ is initialized with the state and action spaces as in the original SSP instance, the horizon length $H$, a confidence parameter $\delta/4$, a set of terminal costs $\hat{c}_f$ and a bound on the expected cost of the optimal policy in the finite-horizon model $9B_\star$. At the beginning of each interval, it takes as input an initial state and outputs a policy to be used throughout the interval. In the end of the interval it receives the trajectory and costs observed through the interval.

Note that while Algorithm 1 may run any finite-horizon regret minimization algorithm, in the analysis we require that $\mathcal{A}$ possesses some properties (that most optimistic algorithms already have) in order to establish our regret bound. We specifically require $\mathcal{A}$ to be an *admissible* algorithm—a model-based optimistic algorithm for regret minimization in finite-horizon MDPs, e.g., UCBVI [Azar et al., 2017] and EULER [Zanette and Brunskill, 2019]. Admissible algorithms are defined formally as follows.

**Definition 1.** A model-based algorithm $\mathcal{A}$ for regret minimization in finite-horizon MDPs is called *admissible* if, when running $\mathcal{A}$ with confidence parameter $\delta$, there is a good event that holds with probability at least $1 - \delta$, under which the following hold:

    (i) $\mathcal{A}$ provides anytime regret guarantees without prior knowledge of the number of episodes, and when the initial state of each episode is arbitrary. The regret bound that $\mathcal{A}$ guarantees for $M$ episodes is denoted by $\widehat{\mathcal{R}}_\mathcal{A}(M)$, for some non-decreasing function $\widehat{\mathcal{R}}_\mathcal{A}$.

    (ii) The policy $\pi^m$ that $\mathcal{A}$ picks in episode $m$ is greedy with respect to an estimate of the optimal policy's $Q$-function.

    (iii) The algorithm's estimate $\underline{J}^m$ of $\widehat{J}^\star$ (the cost-to-go function associated with the optimal finite-horizon policy) is optimistic, i.e., $\underline{J}_h^m(s) \leq \widehat{J}_h^\star(s)$ for every $s \in S$ and $h = 1, \ldots, H + 1$.

(iv) $\mathcal{A}$ computes $\underline{J}^m$ using estimates $\tilde{c}^m, \widetilde{P}^m$ of the cost function $\hat{c}$ and the transition function $\widehat{P}$, respectively. There exists $\omega_{\mathcal{A}}$ which is a function of $H, |S|, |A|$ such that: if state-action pair $(s, a)$ was visited at least $\omega_{\mathcal{A}} \log \frac{MH|S||A|}{\delta}$ times, then $|\tilde{c}_h^m(s, a) - \hat{c}(s, a)| \leq B_\star/H$ and $\|\widetilde{P}^m(\cdot \mid s, a) - \widehat{P}(\cdot \mid s, a)\|_1 \leq 1/(9H)$.

Using an admissible algorithm in Algorithm 1 enables us to bound the total number of intervals, thus ensuring that the agent reaches the goal state in almost every interval. This is because, as $\mathcal{A}$ is optimistic, it will try to avoid the terminal cost (which is suffered in all states except for $g$) by reaching the goal state. In addition, $\mathcal{A}$ will succeed in doing so once it has a good enough estimation of the transition function. Armed with the notion of admissibility, in the sequel we prove the following regret bound for any admissible algorithm $\mathcal{A}$. The proof of Theorem 2.2 is now given by combining Theorem 3.1 with the regret bound of ULCVI in Theorem 2.1.

**Theorem 3.1.** *Let $\mathcal{A}$ be an admissible algorithm for regret minimization in finite-horizon MDPs and denote its regret in $M$ episodes by $\widehat{\mathcal{R}}_{\mathcal{A}}(M)$. Then, running Algorithm 1 with $\mathcal{A}$ ensures that, with probability at least $1 - \delta$,*

$$R_K \leq \widehat{\mathcal{R}}_{\mathcal{A}}\left(4K + 4 \cdot 10^4 |S||A|\omega_{\mathcal{A}} \log \frac{KT_\star |S||A|\omega_{\mathcal{A}}}{\delta}\right)$$
$$+ O\left(\sqrt{(B_\star^2 + B_\star)K \log \frac{KT_\star |S||A|\omega_{\mathcal{A}}}{\delta}} + T_\star \omega_{\mathcal{A}} |S||A| \log^2 \frac{KT_\star |S||A|\omega_{\mathcal{A}}}{\delta}\right),$$

*where $\omega_{\mathcal{A}}$ is a quantity that depends on the algorithm $\mathcal{A}$ and on $|S|, |A|, H$.*

**Remark 5** (Computational complexity). Our reduction directly inherits the computational complexity of the finite-horizon algorithm $\mathcal{A}$ in $M$ episodes, where $M \approx K + poly(|S|, |A|, T_\star)$ by Lemma 4.3. The computational complexity of ULCVI is $O(H|S|^3|A|^2 \log(MH))$, and therefore our optimal regret for SSP is achieved in total computational complexity of $O\left(T_\star |S|^3|A|^2 \log^2 \frac{KT_\star |S||A|}{\delta}\right)$ which is only logarithmic in the number of episodes.

## 3.1 Unknown expected optimal cost $B_\star$

Inspired by techniques for estimation of the SSP-diameter in the adversarial SSP literature [Rosenberg and Mansour, 2020, Chen and Luo, 2021], in Appendix C we show that our reduction does not need to know $B_\star$ in advance, but can instead estimate it on the fly.

We can obtain a reasonable estimate (up to a constant multiplicative factor) of the cost-to-go from state $s$ by running the Bernstein-SSP algorithm of Rosenberg et al. [2020] for regret minimization in SSPs (that does not need to know $B_\star$) with initial state $s$ for roughly $T_\star^2|S|^2|A|$ episodes. Thus, we can apply our reduction while utilizing our first visits to each state in order to estimate its cost-to-go.

We operate in *phases* where each phase ends when some state is visited at least $T_\star^2|S|^2|A|$ times, and all states that were not visited enough are treated as the goal state. Once we reach a poorly visited state, we simply run an episode of the corresponding Bernstein-SSP algorithm. Notice that this comes at a computational cost that is independent of the number of episodes $K$ (since we use Bernstein-SSP for a small number of episodes), and in Appendix C we show that it achieves similar regret bounds with only an additional additive factor of $\tilde{O}(T_\star^3|S|^3|A|)$.

## 4 Regret analysis

In this section we prove Theorem 3.1. Below we give a high-level overview of the proofs and defer the details to Appendix A. We start the analysis with a regret decomposition that states that the SSP regret can be bounded by the sum of two terms: the expected regret of the finite-horizon algorithm, and the deviation of the actual cost in each interval from its expected value. To that end, we use the notations: $M$ for the total number of intervals, $U^m = (s_1^m, a_1^m, \ldots, s_h^m, a_h^m, s_{H+1}^m)$ for the trajectory visited in interval $m$, $C_h^m$ for the cost suffered in step $h$ of interval $m$, $\pi^m$ for the policy chosen by $\mathcal{A}$ for interval $m$, and $\widehat{J}_h^\pi(s)$ for the expected finite-horizon cost when playing policy $\pi$ starting from state $s$ in time step $h$.

**Lemma 4.1.** *For $H = 8T_\star \log(8K)$, we have the following bound on the regret of Algorithm 1:*

$$R_K \leq \widehat{\mathcal{R}}_{\mathcal{A}}(M) + \sum_{m=1}^{M} \left( \sum_{h=1}^{H} C_h^m + \hat{c}_f(s_{H+1}^m) - \widehat{J}_1^{\pi^m}(s_1^m) \right) + B_\star. \tag{1}$$

The bound in Eq. (1) is comprised of two summands and an additional constant. The first summand is an upper bound on the expected finite-horizon regret which we acquire by the admissibility of $\mathcal{A}$ (Definition 1). Note that this bound is in terms of the number of intervals $M$ (i.e., the number of finite-horizon episodes) which is a random variable and not necessarily bounded. In what follows we show that, using the admissibility of $\mathcal{A}$, we can actually bound $M$ by the number of SSP episodes $K$ plus a constant that depends on $\omega_{\mathcal{A}}, |S|, |A|, T_\star$ (but not on $K$). The second summand in Eq. (1) relates to the deviation of the total finite-horizon cost from its expected value.

The proof of Lemma 4.1 builds on two key ideas. The first is that, by setting $H$ to be $O(T_\star \log K)$, we ensure that the expected cost of the optimal policy in the SSP model $\mathcal{M}$ is close to that in the finite-horizon model $\widehat{\mathcal{M}}$. The second idea is that if the agent does not reach the goal state in a certain interval, then she must suffer the terminal cost in the finite-horizon model. Therefore, although in a single episode there may be many intervals in which the agent does not reach the goal state, we can upper bound the cost in these extra intervals in $\mathcal{M}$ by the corresponding terminal costs in $\widehat{\mathcal{M}}$.

Next, we bound the deviation of the actual cost in each interval from its expected value which appears as the second summand in Eq. (1). The bound is due to the following lemma.

**Lemma 4.2.** *Assume that the reduction is performed using an admissible algorithm $\mathcal{A}$. Then, the following holds with probability at least $1 - 3\delta/8$,*

$$\sum_{m=1}^{M} \left( \sum_{h=1}^{H} C_h^m + \hat{c}_f(s_{H+1}^m) - \widehat{J}_1^{\pi^m}(s_1^m) \right) = O\left( \sqrt{(B_\star^2 + B_\star)M \log \frac{M}{\delta}} + H\omega_{\mathcal{A}}|S||A| \log \frac{MKT_\star|S||A|}{\delta} \right).$$

The key observation here relies on the notion of *unknown* state-action pairs – pairs that were not visited at least $\omega_{\mathcal{A}}$ times. After $\omega_{\mathcal{A}}$ visits to some state-action pair $s, a$, we have a reasonable estimate of the next-state distribution $P(\cdot \mid s, a)$ therefore we can show that the expected accumulated cost in an interval until reaching an unknown state-action pair or the goal state is of order $B_\star$. Moreover, the second moment of this cost is of order $B_\star^2 + B_\star$. Thus, using Freedman inequality, we bound the deviation by $\widetilde{O}(\sqrt{(B_\star^2 + B_\star)M})$, plus a cost of $O(H)$ for each "bad" interval in which we do not reach an unknown state-action pair or the goal state (there are roughly $\omega_{\mathcal{A}}|S||A|$ such intervals).

Lastly, we need to bound the number of intervals $M$ to obtain a regret bound in terms of $K$ and not $M$ (notice that $M$ is a random variable that is not bounded a-priori).

**Lemma 4.3.** *Assume that the reduction is performed using an admissible algorithm $\mathcal{A}$. Then, with probability at least $1 - 3\delta/8$, $M \leq 4K + 4 \cdot 10^4 |S||A|\omega_{\mathcal{A}} \log(KT_\star|S||A|\omega_{\mathcal{A}}/\delta)$.*

The proof shows that in every interval there is a constant probability to reach either the goal state or an unknown state-action pair. Leveraging this observation with a concentration inequality, we can bound the number of intervals by $\widetilde{O}(K + \omega_{\mathcal{A}}|S||A|H)$.

We can now prove a bound on the regret of Algorithm 1 using any admissible algorithm $\mathcal{A}$.

*Proof of Theorem 3.1.* The regret bound of $\mathcal{A}$, Lemmas 4.2 and 4.3 all hold with probability at least $1 - \delta$, via a union bound. Using Lemmas 4.1 and 4.2 we can write

$$R_K \leq \widehat{\mathcal{R}}_{\mathcal{A}}(M) + O\left( \sqrt{(B_\star^2 + B_\star)M \log \frac{M}{\delta}} + H\omega_{\mathcal{A}}|S||A| \log \frac{MKT_\star|S||A|}{\delta} \right) + B_\star.$$

Finally, we use Lemma 4.3 to bound $M$ by $4K + 4 \cdot 10^4 |S||A|\omega_{\mathcal{A}} \log(KT_\star|S||A|\omega_{\mathcal{A}}/\delta)$. $\qquad\square$

## 5 ULCVI: an admissible algorithm for finite-horizon MDPs

In this section we present the Upper Lower Confidence Value Iteration algorithm (ULCVI; Algorithm 2) for regret minimization in finite-horizon MDPs. This result holds independently of our SSP

---

**Algorithm 2** UPPER LOWER CONFIDENCE VALUE ITERATION (ULCVI)

---
1: **input:** state space $S$, action space $A$, horizon $H$, confidence parameter $\delta$, terminal costs $\hat{c}_f$ and upper bound on the expected cost of the optimal policy $B_\star$.
2: **initialize:** $n^0(s,a) = 0, n^0(s,a,s') = 0, N^0(s,a) = 0, N^0(s,a,s') = 0 \; \forall(s,a,s') \in S \times A \times S$.
3: **initialize:** $C^0(s,a) = 0, \bar{c}^0(s,a) = 0, \bar{P}^0(s'|s,a) = \mathbb{I}\{s' = s\} \; \forall(s,a,s') \in S \times A \times S$.
4: **initialize:** `PlanningTrigger = true`.
5: **for** $m = 1, 2, \ldots$ **do**
6:    observe initial state $s_1^m$.
7:    **if** `PlanningTrigger = true` **then**
8:        set $n^{m-1}(s,a) \leftarrow N^{m-1}(s,a), n^{m-1}(s,a,s') \leftarrow N^{m-1}(s,a,s') \; \forall(s,a,s')$.
9:        set $\bar{P}^{m-1}(s'|s,a) \leftarrow \frac{n^{m-1}(s,a,s')}{\max\{1,n^{m-1}(s,a)\}}, \bar{c}^{m-1}(s,a) \leftarrow \frac{C^{m-1}(s,a)}{\max\{1,n^{m-1}(s,a)\}} \; \forall(s,a,s')$.
10:        compute $\{\pi_h^m(s)\}_{s,h}$ via OPTIMISTIC-PESSIMISTIC VALUE ITERATION (Algorithm 3).
11:        set `PlanningTrigger` $\leftarrow$ `false`.
12:    **else**
13:        set $n^{m-1}(s,a) \leftarrow n^{m-2}(s,a), n^{m-1}(s,a,s') \leftarrow n^{m-2}(s,a,s') \; \forall(s,a,s')$
14:        set $\bar{P}^{m-1}(s'|s,a) \leftarrow \bar{P}^{m-2}(s'|s,a), \bar{c}^{m-1}(s,a) \leftarrow \bar{c}^{m-2}(s,a) \; \forall(s,a,s')$.
15:        set $\pi_h^m(s) \leftarrow \pi_h^{m-1}(s)$ for all $s \in S$ and $h = 1, \ldots, H$.
16:    **end if**
17:    set $N^m(s,a) \leftarrow N^{m-1}(s,a), N^m(s,a,s') \leftarrow N^{m-1}(s,a,s'), C^m(s,a) \leftarrow C^{m-1}(s,a) \; \forall(s,a,s')$.
18:    **for** $h = 1, \ldots, H$ **do**
19:        pick action $a_h^m = \pi_h^m(s_h^m)$.
20:        suffer cost $C_h^m \sim \hat{c}(s_h^m, a_h^m)$ and observe next state $s_{h+1}^m \sim \widehat{P}(\cdot \mid s_h^m, a_h^m)$.
21:        update visits counters $n^m(s_h^m, a_h^m) \leftarrow n^m(s_h^m, a_h^m) + 1, n^m(s_h^m, a_h^m, s_{h+1}^m) \leftarrow n^m(s_h^m, a_h^m, s_{h+1}^m) + 1$.
22:        update accumulated cost $C^m(s_h^m, a_h^m) \leftarrow C^m(s_h^m, a_h^m) + C_h^m$.
23:        **if** $N^m(s_h^m, a_h^m) \geq 2n^{m-1}(s_h^m, a_h^m)$ **then**
24:            set `PlanningTrigger` $\leftarrow$ `true`.
25:        **end if**
26:    **end for**
27:    Suffer terminal cost $\hat{c}_f(s_{H+1}^m)$.
28: **end for**

---

algorithm. Since the algorithm is similar to previous optimistic algorithms for the finite-horizon setting, e.g., UCBVI [Azar et al., 2017] and ORLC [Dann et al., 2019], we defer the analysis to Appendix B and focus on our technical novelty – bounding the regret in terms of the optimal value function and not the horizon.

In each episode $m$, the ULCVI algorithm maintains an optimistic lower bound $\underline{J}_h^m(s)$ and a pessimistic upper bound $\bar{J}_h^m(s)$ on the cost-to-go function of the optimal policy $J_h^\star(s)$, and acts greedily with respect to the optimistic estimates. These optimistic and pessimistic estimates are computed based on the empirical transition function $\bar{P}^{m-1}(s' \mid s, a)$ and the empirical cost function $\bar{c}^{m-1}(s, a)$ to which we add an exploration bonus $b_c^m(s,a) + b_p^m(s,a)$, where $b_p^m$ handles the approximation error in the transitions and $b_c^m$ handles the approximation error in the costs. The bonuses are defined as follows,

$$b_c^m(s,a) = \sqrt{\frac{2\overline{\text{Var}}_{s,a}^{m-1}(C)L_m}{\max\{1, n^{m-1}(s,a)\}}} + \frac{5L_m}{\max\{1, n^{m-1}(s,a)\}} \tag{2}$$

$$b_p^m(s,a) = \sqrt{\frac{2\text{Var}_{\bar{P}^{m-1}(\cdot|s,a)}(\underline{J}_{h+1}^m)L_m}{\max\{1, n^{m-1}(s,a)\}}} + \frac{62H^3 B_\star^{-1}|S|L_m}{\max\{1, n^{m-1}(s,a)\}} + \frac{B_\star}{16H^2}\mathbb{E}_{\bar{P}^{m-1}(\cdot|s,a)}[\bar{J}_{h+1}^m(s') - \underline{J}_{h+1}^m(s')],$$

where $L_m = 3\log(3|S||A|Hm/\delta)$ is a logarithmic factor and $n^{m-1}(s,a)$ is the number of visits to $(s,a)$ in the first $m-1$ episodes. Furthermore, $\overline{\text{Var}}_{s,a}^{m-1}(C)$ is the empirical variance of the observed costs in $(s,a)$ in the first $m-1$ episodes.[2] Lastly, the term $\text{Var}_{\bar{P}^{m-1}(\cdot|s,a)}(\underline{J}_{h+1}^m)$ is the variance of the next state value $\underline{J}_{h+1}^m$ from state-action pair $(s,a)$, calculated via the empirical transition model, i.e., $\text{Var}_{\bar{P}^{m-1}(\cdot|s,a)}(\underline{J}_{h+1}^m) = \mathbb{E}_{\bar{P}^{m-1}(\cdot|s,a)}[\underline{J}_{h+1}^m(s')^2] - \mathbb{E}_{\bar{P}^{m-1}(\cdot|s,a)}[\underline{J}_{h+1}^m(s')]^2$.

---

[2] The empirical variance of $n$ numbers $a_1, \ldots, a_n$ is defined by $\frac{1}{n}\sum_{i=1}^n \left(a_i - \frac{1}{n}\sum_{j=1}^n a_j\right)^2$.

---

**Algorithm 3** OPTIMISTIC-PESSIMISTIC VALUE ITERATION

---

1: **input:** $n^{m-1}, \bar{P}^{m-1}, \bar{c}^{m-1}, \hat{c}_f, B_\star$.
2: **initialize** $\underline{J}_{H+1}^m(s) = \bar{J}_{H+1}^m(s) = \hat{c}_f(s)$ for all $s \in S$.
3: **for** $h = H, H-1, \dots, 1$ **do**
4:     **for** $s \in S$ **do**
5:         **for** $a \in A$ **do**
6:             set the bonus $b_h^m(s, a) = b_c^m(s, a) + b_p^m(s, a)$ defined in Eq. (2).
7:             compute optimistic and pessimistic Q-functions:

$$\underline{Q}_h^m(s, a) = \bar{c}^{m-1}(s, a) - b_h^m(s, a) + \mathbb{E}_{\bar{P}^{m-1}(\cdot|s,a)}[\underline{J}_{h+1}^m(s')]$$

$$\bar{Q}_h^m(s, a) = \bar{c}^{m-1}(s, a) + b_h^m(s, a) + \mathbb{E}_{\bar{P}^{m-1}(\cdot|s,a)}[\bar{J}_{h+1}^m(s')].$$

8:         **end for**
9:         $\pi_h^m(s) \in \arg\min_{a \in A} \underline{Q}_h^m(s, a)$.
10:         $\underline{J}_h^m(s) = \max\left\{\underline{Q}_h^m(s, \pi_h^m(s)), 0\right\}, \bar{J}_h^m(s) = \min\left\{\bar{Q}_h^m(s, \pi_h^m(s)), H\right\}$.
11:     **end for**
12: **end for**

---

For improved computational complexity, we compute the optimistic policy only in episodes in which the number of visits to some state-action pair was doubled. This ensures that the number of optimistic policy computations grows only logarithmically with the number of episodes, i.e., it is bounded by $3|S||A|\log(MH)$. Since each optimal policy computation costs $O(H|S|^2|A|)$ in the finite-horizon MDP model, our algorithm enjoys a total computational complexity of $O(H|S|^3|A|^2 \log(MH))$.

For clarity, we keep the notation of the finite-horizon MDP as $\widehat{\mathcal{M}} = (S, A, \widehat{P}, H, \hat{c}, \hat{c}_f)$, and let $B_\star = \max_{s,h} \widehat{J}_h^\star(s)$ where $\widehat{J}^\pi$ is the value function of policy $\pi$ (in the case of our SSP reduction this parameter is simply $9B_\star$ by Lemma A.1). This implies that $\hat{c}_f(s) \leq B_\star$ for every $s$, and for simplicity, we assume that $B_\star \leq H$. Thus, the maximal total cost in an episode is bounded by $H + B_\star \leq 2H$. In Appendix B we prove the following high probability regret bound.

**Theorem 5.1.** ULCVI *(Algorithm 2) is admissible with the following guarantees:*

    *(i) With probability at least $1 - \delta$, the regret bound of* ULCVI *is*

$$\widehat{\mathcal{R}}_{\text{ULCVI}}(M) = O\left(\sqrt{(B_\star^2 + B_\star)|S||A|M} \log \frac{MH|S||A|}{\delta} + H^4 B_\star^{-1}|S|^2|A| \log^{3/2} \frac{MH|S||A|}{\delta}\right)$$

    *for any number of episodes $M \geq 1$.*
    *(ii) $\omega_{\text{ULCVI}} = O(H^4 B_\star^{-2}|S|)$.*

Our analysis resembles the one in Efroni et al. [2021], and is adapted to the stationary MDP setting (i.e., the transition function does not depend on the time step $h$), and to the setting where we have costs instead of rewards, and terminal costs (which do not appear in previous work). By the definition of the algorithm and the regret bound in Theorem 5.1, it is clear that properties (i)-(iii) in Definition 1 of admissible algorithms hold. For property (iv), we use standard concentration inequalities and the definition of the bonuses in Eq. (2) in order to show it holds for $\omega_{\text{ULCVI}} = O(H^4 B_\star^{-2}|S|)$.

To obtain a regret bound whose leading term depends on $B_\star$ and not $H$, we start with a standard regret analysis for optimistic algorithms that establishes the regret scales with the square-root of the variance of the value functions of the agent's policies, i.e.,

$$\widehat{\mathcal{R}}_{\text{ULCVI}}(M) \lesssim \sqrt{|S||A|} \sqrt{\sum_{m=1}^M \sum_{h=1}^H \text{Var}_{P(\cdot|s_h^m, a_h^m)}(J_{h+1}^{\pi^m})} + H^4 B_\star^{-1}|S|^2|A|,$$

up to logarithmic factors and lower order terms. This can be further bounded by the second moment of the cumulative cost in each episode as follows,

$$\widehat{\mathcal{R}}_{\text{ULCVI}}(M) \lesssim \sqrt{|S||A|} \sqrt{\sum_{m=1}^M \mathbb{E}\left[\left(\sum_{h=1}^H C_h^m + \hat{c}_f(s_{H+1}^m)\right)^2 \Bigg| \bar{U}^m\right]} + H^4 B_\star^{-1}|S|^2|A|,$$

where $\bar{U}^m$ is the sequence of state-action pairs observed up to episode $m$. Leveraging our techniques for the SSP reduction (but independently), we show that the second moment of the cumulative cost until an unknown state-action pair is reached can be bounded by $O(B_\star^2 + B_\star)$. Therefore, we have at most $\widetilde{O}(H^4 B_\star^{-2}|S|^2|A|)$ episodes in which we bound the second moment trivially by $O(H^2)$, and in the rest of the episodes we can bound it by $O(B_\star^2 + B_\star)$. Together this yields the theorem as follows,

$$\widehat{\mathcal{R}}_{\mathrm{ULCVI}}(M) \lesssim \sqrt{|S||A|}\sqrt{(B_\star^2 + B_\star)M + H^2 \cdot H^4 B_\star^{-2}|S|^2|A|} \lesssim \sqrt{(B_\star^2 + B_\star)|S||A|M} + H^4 B_\star^{-1}|S|^2|A|.$$

## Acknowledgements

This project has received funding from the European Research Council (ERC) under the European Union'sHorizon 2020 research and innovation program (grant agreement No. 882396), by the Israel Science Foundation(grant number 993/17), Tel Aviv University Center for AI and Data Science (TAD), and the Yandex Initiative for Machine Learning at Tel Aviv University

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
