# A  Proofs for Section 4

## A.1  Proof of Lemma 4.1

In this section we relate the SSP regret and the finite-horizon regret, which relies on Lemmas A.1 and A.2 below that compare the cost-to-go function in the SSP $\mathcal{M}$ to the value function in the finite-horizon $\widehat{\mathcal{M}}$. To that end, we define a cost-to-go function with respect to the finite-horizon MDP $\widehat{\mathcal{M}}$ as: $\widehat{J}_h^\pi(s) = \mathbb{E}\left[\sum_{h'=h}^H c(s_{h'}, a_{h'}) \mid s_h = s\right]$, for any deterministic finite-horizon policy $\pi : S \times [H] \mapsto A$.

**Lemma A.1.** *Let $\pi$ be a stationary policy. For every $s \in \widehat{S}$ and $h = 1, \ldots, H+1$ it holds that*

$$\widehat{J}_h^\pi(s) \leq J^\pi(s) + 8B_\star \mathbb{P}[s_{H+1} \neq g \mid s_h = s, \widehat{P}, \pi].$$

*Proof.*

$$\widehat{J}_h^\pi(s) = \sum_{h'=h}^H \sum_{s' \in \widehat{S}} \mathbb{P}[s_{h'} = s' \mid s_h = s, \widehat{P}, \pi]\, \hat{c}(s', \pi(s')) + \sum_{s' \in \widehat{S}} \mathbb{P}[s_{H+1} = s' \mid s_h = s, \widehat{P}, \pi]\, \hat{c}_f(s')$$

$$= \sum_{h'=h}^H \sum_{s' \in S} \mathbb{P}[s_{h'} = s' \mid s_h = s, P, \pi]\, c(s', \pi(s')) + 8B_\star\, \mathbb{P}[s_{H+1} \neq g \mid s_h = s, \widehat{P}, \pi]$$

$$\leq \sum_{h'=h}^\infty \sum_{s' \in S} \mathbb{P}[s_{h'} = s' \mid s_h = s, P, \pi]\, c(s', \pi(s')) + 8B_\star\, \mathbb{P}[s_{H+1} \neq g \mid s_h = s, \widehat{P}, \pi]$$

$$= J^\pi(s) + 8B_\star\, \mathbb{P}[s_{H+1} \neq g \mid s_h = s, \widehat{P}, \pi]. \qquad \square$$

**Lemma A.2.** *For every $s \in \widehat{S}$, it holds that $J^{\pi^\star}(s) \geq \widehat{J}_1^{\pi^\star}(s) - \frac{B_\star}{K}$.*

*Proof.* The probability that $\pi^\star$ does not reach the goal in $H$ steps is at most $1/(8K)$ due to Chen et al. [2020, Lemma 7]. Plugging that into Lemma A.1 yields the desired result. $\square$

*Proof of Lemma 4.1.* Consider the first interval of the first episode. If it ends in the goal state then

$$\sum_{i=1}^{I^1} C_i^1 = \sum_{h=1}^H C_h^1 + \hat{c}_f(g) = \sum_{h=1}^H C_h^1 + \hat{c}_f(s_{H+1}^1).$$

If the agent did not reach $g$ in the first interval, then the agent also suffered the $8B_\star$ terminal cost and thus

$$\sum_{i=1}^{I^1} C_i^1 = \sum_{h=1}^H C_h^1 + \hat{c}_f(s_{H+1}^1) + \sum_{i=H+1}^{I^1} C_i^1 - \hat{c}_f(s_{H+1}^1)$$

$$= \sum_{h=1}^H C_h^1 + \hat{c}_f(s_{H+1}^1) + \sum_{i=H+1}^{I^1} C_i^1 - 8B_\star$$

$$\leq \sum_{h=1}^H C_h^1 + \hat{c}_f(s_{H+1}^1) + \sum_{i=H+1}^{I^1} C_i^1 - \widehat{J}_1^{\pi^\star}(s_{H+1}^1),$$

where the last inequality follows by combining Lemma A.2 with our assumption that $J^{\pi^\star}(s) \leq B_\star$.

Repeating this argument iteratively we get, for every episode $k$,

$$\sum_{i=1}^{I^k} C_i^k - J^{\pi^\star}(s_{\text{init}}) \leq \sum_{i=1}^{I^k} C_i^k - \widehat{J}_1^{\pi^\star}(s_1^m) + \frac{B_\star}{K}$$

$$\leq \sum_{m \in M_k} \sum_{h=1}^H C_h^m + \hat{c}_f(s_{H+1}^m) - \widehat{J}_1^{\pi^\star}(s_1^m) + \frac{B_\star}{K}$$

$$= \sum_{m \in M_k} \left( \sum_{h=1}^H C_h^m + \hat{c}_f(s_{H+1}^m) - \widehat{J}^{\pi^m}(s_1^m) \right) + \sum_{m \in M_k} \left( \widehat{J}^{\pi^m}(s_1^m) - \widehat{J}_1^{\pi^\star}(s_1^m) \right) + \frac{B_\star}{K},$$

where $M_k$ is the set of intervals that are contained in episode $k$, and the first inequality follows from Lemma A.2. Summing over all episodes obtains

$$R_K \le \sum_{m=1}^{M} \left( \sum_{h=1}^{H} C_h^m + \hat{c}_f(s_{H+1}^m) - \widehat{J}^{\pi^m}(s_1^m) \right) + \sum_{m=1}^{M} \left( \widehat{J}^{\pi^m}(s_1^m) - \widehat{J}_1^{\pi^\star}(s_1^m) \right) + \frac{B_\star}{K}.$$

Notice that the second summand in the bound above is exactly the expected finite-horizon regret over the $M$ intervals. We finish the proof of the lemma by using the regret guarantees of $\mathcal{A}$ (Definition 1).

$\square$

## A.2 Proof of Lemma 4.2

In this section we bound the deviation of the actual cost in each interval from its expected value. To do that, we apply Lemma A.3 below to bound the second moment of the cumulative cost in an interval up until an unknown state-action pair or the goal state were reached. Here $\bar{U}^m$ denotes the union of all information prior to the $m^{th}$ interval together with the first state of the $m^{th}$ interval (more formally, $\{\bar{U}^m\}_{m \ge 1}$ is a filtration). Moreover, we denote by $h_m$ the last time step before an unknown state-action pair or the goal state were reached in interval $m$ (or $H$ if they were not reached).

**Lemma A.3.** *Let m be an interval and assume that the reduction is performed using an admissible algorithm $\mathcal{A}$. If the good event of $\mathcal{A}$ holds until the beginning of interval m, then the agent reaches the goal state or an unknown state-action pair with probability at least $\frac{1}{2}$. Moreover, denote by $C^m = \sum_{h=1}^{h_m} C_h^m + \hat{c}_f(s_{H+1}^m)\mathbb{I}\{h_m = H\}$ the cumulative cost in the interval until time $h_m$. Then, $\mathbb{E}[(C^m)^2 \mid \bar{U}^m] \le 2 \cdot 10^5 B_\star^2 + 4B_\star$.*

*Proof.* The result is given by bounding the total expected cost suffered by the agent in another MDP (defined below) where all unknown state-action pairs are contracted with the goal state. The cost in this MDP is exactly $C^m$ by definition.

Let $\pi^m$ be the optimistic policy chosen by the algorithm for interval $m$. Consider the following finite-horizon MDP $\widehat{\mathcal{M}}^m = (\widehat{S}, A, \widehat{P}^m, H, \hat{c}, \hat{c}_f)$ that contracts unknown state-action pairs with the goal:

$$\widehat{P}_h^m(s' \mid s, a) = \begin{cases} 0, & (s', \pi_{h+1}^m(s')) \text{ is unknown;} \\ P(s' \mid s, a), & s' \ne g \text{ and } (s', \pi_{h+1}^m(s')) \text{ is known;} \\ 1 - \sum_{s'' \in \widehat{S}\setminus\{g\}} \widehat{P}_h^m(s'' \mid s, a), & s' = g. \end{cases}$$

Denote by $J^m$ the cost-to-go function of $\pi^m$ in the finite-horizon MDP $\widehat{\mathcal{M}}^m$. Further, let $\widetilde{P}'^m$ be the transition function induced by $\widetilde{P}^m$ in the MDP $\widehat{\mathcal{M}}'^m$ similarly to $\widehat{P}^m$, and $\widetilde{J}^m$ the cost-to-go function of $\pi^m$ with respect to $\widetilde{P}'^m$ (and with cost function $\tilde{c}^m$). Notice that $\pi^m$ can only reach the goal state quicker in $\widehat{\mathcal{M}}^m$ than in $\widehat{\mathcal{M}}$, so that $\widetilde{J}_h^m(s) \le \underline{J}_h^m(s) \le \widehat{J}_h^{\pi^\star}(s)$ for any $s \in \widehat{S}$. By the value difference lemma (see, e.g., Shani et al., 2020), for every $s, h$ such that $(s, \pi_h^m(s))$ is known,

$$J_h^m(s) = \widetilde{J}_h^m(s) + \sum_{h'=h}^{H} \mathbb{E}\left[ \hat{c}(s_{h'}, a_{h'}) - \tilde{c}_{h'}^m(s_{h'}, a_{h'}) \mid s_h = s, \widehat{P}^m, \pi^m \right]$$

$$+ \sum_{h'=h}^{H} \mathbb{E}\left[ \left( \widehat{P}_{h'}^m(\cdot \mid s_{h'}, a_{h'}) - \widetilde{P}_{h'}'^m(\cdot \mid s_{h'}, a_{h'}) \right) \cdot \widetilde{J}^m \mid s_h = s, \widehat{P}^m, \pi^m \right]$$

$$\le \widetilde{J}_h^m(s) + H \max_{\substack{(s, \pi_{h'}^m(s)) \\ \text{known}}} |c(s, \pi_{h'}^m(s)) - \tilde{c}_{h'}^m(s, \pi_{h'}^m(s))| + H\|\widetilde{J}^m\|_\infty \max_{\substack{(s, \pi_{h'}^m(s)) \\ \text{known}}} \|\widehat{P}_{h'}^m(\cdot|s, \pi_{h'}^m(s)) - \widetilde{P}_{h'}'^m(\cdot|s, \pi_{h'}^m(s))\|_1$$

$$\overset{(a)}{\le} \widehat{J}_h^{\pi^\star}(s) + H \max_{\substack{(s, \pi_{h'}^m(s)) \\ \text{known}}} |c(s, \pi_{h'}^m(s)) - \tilde{c}_{h'}^m(s, \pi_{h'}^m(s))|$$

$$+ H\|\widehat{J}_h^{\pi^\star}(s)\|_\infty \max_{\substack{(s, \pi_{h'}^m(s)) \\ \text{known}}} \|\widehat{P}(\cdot|s, \pi_{h'}^m(s)) - \widetilde{P}^m(\cdot|s, \pi_{h'}^m(s))\|_1$$

$$\le \widehat{J}_h^{\pi^\star}(s) + H \max_{\substack{(s, \pi_{h'}^m(s)) \\ \text{known}}} |c(s, \pi_{h'}^m(s)) - \tilde{c}_{h'}^m(s, \pi_{h'}^m(s))| + 9HB_\star \max_{\substack{(s, \pi_{h'}^m(s)) \\ \text{known}}} \|\widehat{P}(\cdot|s, \pi_{h'}^m(s)) - \widetilde{P}^m(\cdot|s, \pi_{h'}^m(s))\|_1,$$

where the last inequality follows by optimism and since $\widehat{J}_h^{\pi^\star}(s) \leq 9B_\star$ (Lemma A.1), and (a) follows because

$$\|\widehat{P}_h^m(\cdot|s,a) - \widetilde{P}_h'^m(\cdot|s,a)\|_1 = \sum_{\substack{(s',\pi_{h+1}^m(s')) \\ \text{known}}} |\widehat{P}_h^m(s'|s,a) - \widetilde{P}_h'^m(s'|s,a)| + |\widehat{P}_h^m(g|s,a) - \widetilde{P}_h'^m(g|s,a)|$$

$$= \sum_{\substack{(s',\pi_{h+1}^m(s')) \\ \text{known}}} |\widehat{P}(s'|s,a) - \widetilde{P}^m(s'|s,a)| + \left| \sum_{\substack{(s',\pi_{h+1}^m(s')) \\ \text{unknown}}} \widehat{P}(s'|s,a) + \widehat{P}(g|s,a) - \widetilde{P}^m(s'|s,a) - \widetilde{P}^m(g|s,a) \right|$$

$$\leq \|\widehat{P}(\cdot|s,a) - \widetilde{P}^m(\cdot|s,a)\|_1 .$$

Thus $J_h^m(s) \leq \widehat{J}_h^{\pi^\star}(s) + 2B_\star$ since the number of visits to each known state-action pair is at least $\omega_{\mathcal{A}} \log \frac{MH|S||A|}{\delta}$ and by property (iv) of admissible algorithms (Definition 1). Also note that $J_h^m(s) \leq 11B_\star$ by Lemma A.1, and for $h = 1$ in particular we use Lemma A.2 to obtain $J_1^m(s) \leq 4B_\star$.

By Markov inequality, the probability that the agent suffers a cost of more than $8B_\star$ in $\widehat{\mathcal{M}}^m$ is at most $\frac{1}{2}$. Notice that all costs are non-negative and there is a terminal cost of $8B_\star$ in all states but the goal, therefore the agent cannot suffer a cost of less than $8B_\star$ unless she reaches the goal. So the probability to reach the goal is at least $\frac{1}{2}$. Moreover, note that the probability to reach the goal in $\widehat{\mathcal{M}}^m$ is equal to the probability to reach the goal or an unknown state-action pair in $\widehat{\mathcal{M}}$.

Similarly, we notice that $\mathbb{E}[(C^m)^2 \mid \bar{U}^m] = \mathbb{E}[(\widehat{C})^2]$, where $\widehat{C}$ is the cumulative cost in $\widehat{\mathcal{M}}^m$, and we override notation by denoting $\widehat{C} = \sum_{h=1}^{H} C_h + \hat{c}_f(s_{H+1})$. We have that,

$$\mathbb{E}[(\widehat{C})^2] = \mathbb{E}\left[ \left( \sum_{h=1}^{H} C_h + \hat{c}_f(s_{H+1}) \right)^2 \right]$$

$$= \mathbb{E}\left[ \left( \sum_{h=1}^{H-1} C_h + \hat{c}(s_H, a_H) + \hat{c}_f(s_{H+1}) \right)^2 \right]$$

$$+ 2\mathbb{E}\left[ \left( \sum_{h=1}^{H-1} C_h + \hat{c}(s_H, a_H) + \hat{c}_f(s_{H+1}) \right) (C_H - \hat{c}(s_H, a_H)) \right] + \mathbb{E}[(C_H - \hat{c}(s_H, a_H))^2].$$

The second summand is zero since the realization of $C_H$ is independent of all other randomness given $s_H$. Also, since $C_H \in [0, 1]$, the third summand satisfies

$$\mathbb{E}[(C_H - \hat{c}(s_H, a_H))^2] \leq \mathbb{E}[(C_H)^2] \leq \mathbb{E}[C_H] = \mathbb{E}[\hat{c}(s_H, a_H)].$$

Thus we arrived at

$$\mathbb{E}[(\widehat{C})^2] \leq \mathbb{E}\left[ \left( \sum_{h=1}^{H-1} C_h + \hat{c}(s_H, a_H) + \hat{c}_f(s_{H+1}) \right)^2 \right] + \mathbb{E}[\hat{c}(s_H, a_H)],$$

and iterating this argument yields

$$\mathbb{E}[(\widehat{C})^2] \leq \mathbb{E}\left[ \left( \sum_{h=1}^{H} \hat{c}(s_h, a_h) + \hat{c}_f(s_{H+1}) \right)^2 \right] + \mathbb{E}\left[ \sum_{h=1}^{H} \hat{c}(s_h, a_h) \right].$$

Here, the second summand equals $J_1^m(s_1)$ which is at most $4B_\star$.

Next, for the first summand, we split the time steps into $Q$ blocks as follows. We denote by $t_1$ the first time step in which we accumulated a total cost of at least $11B_\star$ (or $H + 1$ if it did not occur), by $t_2$ the first time step in which we accumulated a total cost of at least $11B_\star$ after $t_1$, and so on up until $t_Q = H + 1$. Then, the first block consists of time steps $t_0 = 1, \ldots, t_1 - 1$, the second block consists of time steps $t_1, \ldots, t_2 - 1$, and so on. Since $J_h^m(s) \leq 11B_\star$ we must have $\hat{c}(s_h, a_h) \leq 11B_\star$ for all

$h = 1, \ldots, H$ and thus in every such block the total cost is between $11B_\star$ and $22B_\star$. Thus,

$$\mathbb{E}\left[\left(\sum_{h=1}^{H} \hat{c}(s_h, a_h) + \hat{c}_f(s_{H+1})\right)^2\right] \geq \mathbb{E}\left[\sum_{h=1}^{H} \hat{c}(s_h, a_h) + \hat{c}_f(s_{H+1})\right]^2$$

$$= \mathbb{E}\left[\sum_{i=0}^{Q-1} \sum_{h=t_i}^{t_{i+1}-1} \hat{c}(s_h, a_h) + \hat{c}_f(s_{H+1})\right]^2$$

$$\geq \mathbb{E}[11B_\star Q]^2 = 121B_\star^2 \mathbb{E}[Q]^2,$$

by Jensen's inequality. On the other hand,

$$\mathbb{E}\left[\left(\sum_{h=1}^{H} \hat{c}(s_h, a_h) + \hat{c}_f(s_{H+1})\right)^2\right] = \mathbb{E}\left[\left(\sum_{h=1}^{H} \hat{c}(s_h, a_h) + \hat{c}_f(s_{H+1}) - J_1^m(s_1) + J_1^m(s_1)\right)^2\right]$$

$$\leq 2\mathbb{E}\left[\left(\sum_{h=1}^{H} \hat{c}(s_h, a_h) + \hat{c}_f(s_{H+1}) - J_1^m(s_1)\right)^2\right] + 2J_1^m(s_1)^2$$

$$\leq 2\mathbb{E}\left[\left(\sum_{i=0}^{Q-1} \sum_{h=t_i}^{t_{i+1}-1} \hat{c}(s_h, a_h) - J_{t_i}^m(s_{t_i}) + J_{t_{i+1}}^m(s_{t_{i+1}})\right)^2\right] + 32B_\star^2$$

$$\overset{(a)}{=} 4\mathbb{E}\left[\sum_{i=0}^{Q-1} \left(\sum_{h=t_i}^{t_{i+1}-1} \hat{c}(s_h, a_h) - J_{t_i}^m(s_{t_i}) + J_{t_{i+1}}^m(s_{t_{i+1}})\right)^2\right] + 32B_\star^2$$

$$\leq 4\mathbb{E}[Q \cdot (33B_\star)^2] + 32B_\star^2 \leq 4356B_\star^2 \mathbb{E}[Q] + 32B_\star^2.$$

For (a) we used the fact that $\mathbb{E}[\sum_{h=t_i}^{t_{i+1}-1} \hat{c}(s_h, a_h) - J_{t_i}(s_{t_i}) + J_{t_{i+1}}(s_{t_{i+1}})] = 0$ using the Bellman optimality equations and conditioned on all past randomness up until time $t_i$, and the fact that $t_{i+1}$ is a (bounded) stopping time by the optional stopping theorem, in the following manner,

$$\mathbb{E}\left[\sum_{h=t_i}^{t_{i+1}-1} \hat{c}(s_h, a_h) - J_{t_i}^m(s_{t_i}) + J_{t_{i+1}}^m(s_{t_{i+1}})\right] = \mathbb{E}\left[\sum_{h=t_i}^{t_{i+1}-1} \hat{c}(s_h, a_h) - J_h^m(s_h) + J_{h+1}^m(s_{h+1})\right]$$

$$= \mathbb{E}\left[\sum_{h=t_i}^{t_{i+1}-1} \mathbb{E}\left[\hat{c}(s_h, a_h) - J_h^m(s_h) + J_{h+1}^m(s_{h+1}) \mid s_1, \ldots, s_h\right]\right]$$

$$= \mathbb{E}\left[\sum_{h=t_i}^{t_{i+1}-1} \hat{c}(s_h, a_h) + \mathbb{E}\left[J_{h+1}^m(s_{h+1}) \mid s_h\right] - J_h^m(s_h)\right] = 0.$$

Thus, we have $121B_\star^2 \mathbb{E}[Q]^2 \leq 4356B_\star^2 \mathbb{E}[Q] + 32B_\star^2$, and solving for $\mathbb{E}[Q]$ we obtain $\mathbb{E}[Q] \leq 37$, so

$$\mathbb{E}\left[\left(\sum_{h=1}^{H} \hat{c}(s_h, a_h) + \hat{c}_f(s_{H+1})\right)^2\right] \leq 2 \cdot 10^5 B_\star^2,$$

and therefore

$$\mathbb{E}[(\widehat{C})^2] \leq \mathbb{E}\left[\left(\sum_{h=1}^{H} \hat{c}(s_h, a_h) + \hat{c}_f(s_{H+1})\right)^2\right] + \mathbb{E}\left[\sum_{h=1}^{H} \hat{c}(s_h, a_h)\right] \leq 2 \cdot 10^5 B_\star^2 + 4B_\star. \qquad \square$$

*Proof of Lemma 4.2.* Recall that $h_m$ is the last time step before an unknown state-action pair or the goal state were reached (or $H$ if they were not reached) in interval $m$, and let $G^m$ be the event that the good event of algorithm $\mathcal{A}$ holds up to the beginning of interval $m$. We start by decomposing the sum as follows

$$\sum_{m=1}^{M} \left(\sum_{h=1}^{H} C_h^m + \hat{c}_f(s_{H+1}^m) - \widehat{J}_1^{\pi^m}(s_1^m)\right) \mathbb{I}\{G^m\} = \sum_{m=1}^{M} \left(\sum_{h=1}^{h_m} C_h^m + c_f(s_{H+1}^m)\mathbb{I}\{h_m = H\} - \widehat{J}_1^{\pi^m}(s_1^m)\right) \mathbb{I}\{G^m\}$$

$$+ \sum_{m=1}^{M} \left(\sum_{h=h_m+1}^{H} C_h^m + \hat{c}_f(s_{H+1}^m)\mathbb{I}\{h_m \neq H\}\right) \mathbb{I}\{G^m\}.$$

The second term is trivially bounded by $(H + 8B_\star)|S||A|\omega_{\mathcal{A}} \log \frac{MH|S||A|}{\delta}$ since every state-action pair becomes known after $\omega_{\mathcal{A}} \log \frac{MH|S||A|}{\delta}$ visits. Next, since

$$\mathbb{E}\left[\left(\sum_{h=1}^{h_m} C_h^m + c_f(s_{H+1}^m)\mathbb{I}\{h_m = H\}\right)\mathbb{I}\{G^m\} \,\middle|\, \bar{U}^m\right] = \mathbb{E}\left[\sum_{h=1}^{h_m} C_h^m + c_f(s_{H+1}^m)\mathbb{I}\{h_m = H\} \,\middle|\, \bar{U}^m\right]\mathbb{I}\{G^m\}$$
$$\leq \widehat{J}_1^{\pi^m}(s_1^m)\mathbb{I}\{G^m\},$$

the first term is bounded by $\sum_{m=1}^M X^m$ where

$$X^m = \left(\sum_{h=1}^{h_m} C_h^m + c_f(s_{H+1}^m)\mathbb{I}\{h_m = H\} - \mathbb{E}\left[\sum_{h=1}^{h_m} C_h^m + c_f(s_{H+1}^m)\mathbb{I}\{h_m = H\} \,\middle|\, \bar{U}^m\right]\right)\mathbb{I}\{G^m\}$$

is a martingale difference sequence bounded by $H + 8B_\star$ with probability 1. For any fixed $M = m$, by Freedman's inequality (Lemma E.1, we have with probability at least $1 - \frac{\delta}{8m(m+1)}$,

$$\sum_{m'=1}^m X^{m'} \leq \eta \sum_{m'=1}^m \mathbb{E}[(X^{m'})^2 \mid \bar{U}^{m'}] + \frac{\log(8m(m+1)/\delta)}{\eta}$$

for any $\eta \in (0, 1/(H + 8B_\star))$. By Lemma A.3, for some universal constant $\alpha > 0$, that

$$\sum_{m'=1}^m \mathbb{E}[(X^{m'})^2 \mid \bar{U}^{m'}] \leq \alpha m(B_\star^2 + B_\star),$$

and setting $\eta = \min\left\{\sqrt{\frac{\log(8m(m+1)/\delta)}{(B_\star^2 + B_\star)m}}, \frac{1}{H+8B_\star}\right\}$ obtains

$$\sum_{m'=1}^m X^{m'} \leq O\left(\sqrt{(B_\star^2 + B_\star)m \log \frac{m}{\delta}} + (H + B_\star)\log \frac{m}{\delta}\right).$$

Taking a union bound on all values of $m = 1, 2, \ldots$ that the inequality above holds for all such values of $m$ simultaneously with probability at least $1 - \delta/8$. In particular, with probability at least $1 - \delta/8$, we have

$$\sum_{m=1}^M X^m \leq O\left(\sqrt{(B_\star^2 + B_\star)M \log \frac{M}{\delta}} + (H + B_\star)\log \frac{M}{\delta}\right).$$

The proof is concluded via a union bound—both Freedman inequality and the good event of $\mathcal{A}$ hold with probability at least $1 - \frac{3}{8}\delta$, and this implies that $\mathbb{I}\{G^m\} = 1$ for every $m$. □

### A.3   Proof of Lemma 4.3

In this section we bound the number of intervals $M$ with high probability for any admissible algorithm. To that end, we first define the notion of unknown state-action pairs. A state-action pair is defined as *unknown* if the number of times it was visited is at most $\omega_{\mathcal{A}} \log \frac{MH|S||A|}{\delta}$ (and otherwise *known*).

*Proof of Lemma 4.3.* Let $G^m$ be the event that the good event of algorithm $\mathcal{A}$ holds up to the beginning of interval $m$, and define $X^m$ to be 1 if an unknown state-action pair or the goal state were reached during interval $m$ (and 0 otherwise). Notice that $\mathbb{E}[X^m\mathbb{I}\{G^m\} \mid \bar{U}^m] = \mathbb{E}[X^m \mid \bar{U}^m]\mathbb{I}\{G^m\} \geq \mathbb{I}\{G^m\}/2$ by Lemma A.3. Moreover, note that every state-action pair becomes known after $\omega_{\mathcal{A}} \log \frac{MH|S||A|}{\delta}$ visits and therefore $\sum_{m=1}^M X^m\mathbb{I}\{G^m\} \leq \sum_{m=1}^M X^m \leq K + |S||A|\omega_{\mathcal{A}} \log \frac{MH|S||A|}{\delta}$. By Lemma E.2, which is a consequence of Freedman's inequality for bounded positive random variables, we have with probability at least $1 - \frac{\delta}{8}$ for all $M \geq 1$ simultaneously

$$\sum_{m=1}^M \mathbb{E}[X^m\mathbb{I}\{G^m\} \mid \bar{U}^m] \leq 2\sum_{m=1}^M X^m\mathbb{I}\{G^m\} + 108 \log \frac{M}{\delta} \leq 2K + 110|S||A|\omega_{\mathcal{A}} \log \frac{MH|S||A|}{\delta}.$$

Using a union bound, this inequality and the good event of $\mathcal{A}$ both hold with probability at least $1 - \frac{3}{8}\delta$. Then, $\mathbb{I}\{G^m\} = 1$ for all $m$, and therefore

$$\frac{M}{2} \leq 2K + 110|S||A|\omega_{\mathcal{A}} \log \frac{MH|S||A|}{\delta}.$$

Using the fact that $x \leq a \log(bx) + c \rightarrow x \leq 6a \log(abc) + c$ for $a, b, c \geq 1$, this implies

$$M \leq 4K + 4 \cdot 10^4 |S||A|\omega_{\mathcal{A}} \log \frac{KT_\star|S||A|\omega_{\mathcal{A}}}{\delta}. \qquad \square$$

# B  Proofs for Section 5

Since all the proofs in this section refer to the finite-horizon setting (without a connection to SSP), we use the simpler notations $\mathcal{M} = (S, A, P, H, c, c_f)$ for the MDP, $J_h^\pi(s)$ for the value function of policy $\pi$, and $B_\star \geq \max_{s,h} J_h^\star(s)$ for the upper bound on the value function of the optimal policy.

We define a state-action pair $(s, a)$ to be *known* if it was visited at least $\alpha H^4 B_\star^{-2}|S|$ times (for some universal constant $\alpha > 0$ to be determined later), and otherwise *unknown*. In addition, we denote by $h_m$ the last time step before an unknown state-action pair was reached (or $H$ if they were not reached).

## B.1  The good event, optimism and pessimism

Throughout this section we use the notation $a \vee 1$ defined as $\max\{a, 1\}$. In addition, we define the logarithmic factor $L_m = 3\log(6|S||A|Hm/\delta)$. Define the following events:

$$E^c(m) = \left\{ \forall(s,a) : |\bar{c}^{m-1}(s,a) - c(s,a)| \leq b_c^m(s,a) \right\}$$

$$E^{cv}(m) = \left\{ \forall(s,a) : \left| \sqrt{\overline{\text{Var}}_{s,a}^{m-1}(C)} - \sqrt{\text{Var}_{s,a}(c)} \right| \leq \sqrt{\frac{12L_m}{n^{m-1}(s,a) \vee 1}} \right\}$$

$$E^p(m) = \left\{ \forall(s,a,s') : |P(s'|s,a) - \bar{P}^{m-1}(s'|s,a)| \leq \sqrt{\frac{2P(s'|s,a)L_m}{n^{m-1}(s,a) \vee 1}} + \frac{2L_m}{n^{m-1}(s,a) \vee 1} \right\}$$

$$E^{pv1}(m) = \left\{ \forall(s,a,h) : \left| \left(\bar{P}^{m-1}(\cdot|s,a) - P(\cdot|s,a)\right) \cdot J_{h+1}^* \right| \leq \sqrt{\frac{2\text{Var}_{P(\cdot|s,a)}(J_{h+1}^*)L_m}{n^{m-1}(s,a) \vee 1}} + \frac{5B_\star L_m}{n^{m-1}(s,a) \vee 1} \right\}$$

$$E^{pv2}(m) = \left\{ \forall(s,a,h) : \left| \sqrt{\text{Var}_{P(\cdot|s,a)}(J_{h+1}^*)} - \sqrt{\text{Var}_{\bar{P}^{m-1}(\cdot|s,a)}(J_{h+1}^*)} \right| \leq \sqrt{\frac{12B_\star^2 L_m}{n^{m-1}(s,a) \vee 1}} \right\}$$

For brevity, we denote $b_{pv1,h}^m(s,a) = \sqrt{\frac{2\text{Var}_{P(\cdot|s,a)}(J_{h+1}^*)L_m}{n^{m-1}(s,a) \vee 1}} + \frac{5B_\star L_m}{n^{m-1}(s,a) \vee 1}$. This good event, which is the intersection of the above events, is the one used in Efroni et al. [2021]. The following lemma establishes that the good event holds with high probability. The proof is supplied in Efroni et al. [2021, Lemma 13] by applying standard concentration results.

**Lemma B.1** (The First Good Event). *Let* $\mathbb{G}_1 = \cap_{m \geq 1} E^c(m) \cap_{m \geq 1} E^{cv}(m) \cap_{m \geq 1} E^p(m) \cap_{m \geq 1} E^{pv1}(m) \cap_{m \geq 1} E^{pv2}(m)$ *be the basic good event. It holds that* $\mathbb{P}(\mathbb{G}_1) \geq 1 - \frac{1}{4}\delta$.

Under the first good event, we can prove that the value is optimistic using standard techniques.

**Lemma B.2** (Upper Value Function is Optimistic, Lower Value Function is Pessimistic). *Conditioned on the first good event* $\mathbb{G}_1$*, it holds that* $\underline{J}_h^m(s) \leq J_h^*(s) \leq J_h^{\pi^m}(s) \leq \bar{J}_h^m(s)$ *for every* $m = 1, 2, \ldots, s \in S$ *and* $h = 1, \ldots, H + 1$.

*Proof.* Since $J_h^*(s) \leq J_h^\pi(s)$ for any policy $\pi$, we only need to prove the leftmost and rightmost inequalities of the claim. We prove this result via induction.

**Base case, the claim holds for** $h = H + 1$. Since we assume the terminal costs are known, for any $s \in S$,

$$\underline{J}_{H+1}^m(s) = J_{H+1}^*(s) = J_{H+1}^{\pi^m}(s) = \bar{J}_{H+1}^m(s) = c_f(s).$$

**Induction step, prove for** $h \in [H]$ **assuming the claim holds for all** $h + 1 \leq h' \leq H + 1$.

**Leftmost inequality, optimism.** Let $a^*(s) \in \arg\min_{a \in A} Q_h^*(s,a)$, then

$$J_h^*(s) - \underline{J}_h^m(s) = Q_h^*(s, a^*(s)) - \max\left\{ \min_{a \in A} \underline{Q}_h^m(s,a), 0 \right\}. \tag{3}$$

Assume that $\min_a \bar{Q}_h^m(s,a) > 0$ (otherwise, the inequality is satisfied). Then,

$$
\begin{aligned}
(3) &\geq Q_h^*(s, a^*(s)) - \underline{Q}_h^m(s, a^*(s)) \\
&= c(s, a^*(s)) - \bar{c}^{m-1}(s, a^*(s)) + b_c^m(s, a^*(s)) + b_p^m(s, a^*(s)) \\
&\quad + (P - \bar{P}^{m-1})(\cdot \mid s, a^*(s)) \cdot J_{h+1}^* + \mathbb{E}_{\bar{P}^{m-1}(\cdot \mid s, a^*(s))}[\underbrace{J_{h+1}^*(s') - \underline{J}_{h+1}^m(s')}_{\geq 0 \text{ Induction hypothesis}}] \\
&\geq -b_{pv1,h}^m(s, a^*(s)) + b_p^m(s, a^*(s)),
\end{aligned}
\tag{4}
$$

where the last relation holds since the events $\cap_m E^{pv1}(m)$ and $\cap_m E^c(m)$ hold. We now analyze this term.

$$
\begin{aligned}
(4) &= -b_{pv1,h}^m(s, a^*(s)) + b_p^m(s, a^*(s)) \\
&\stackrel{(a)}{\geq} -\sqrt{\frac{2\mathrm{Var}_{P(\cdot\mid s, a^*(s))}(J_{h+1}^*)L_m}{n^{m-1}(s, a^*(s)) \vee 1}} - \frac{5B_\star L_m}{n^{m-1}(s, a^*(s)) \vee 1} \\
&\quad + \sqrt{\frac{2\mathrm{Var}_{\bar{P}^{m-1}(\cdot\mid s, a^*(s))}(\underline{J}_{h+1}^m)L_m}{n^{m-1}(s, a^*(s)) \vee 1}} + \frac{17H^3 B_\star^{-1} L_m}{n^{m-1}(s, a^*(s)) \vee 1} + \frac{B_\star}{16H^2}\mathbb{E}_{\bar{P}^{m-1}(\cdot\mid s, a)}\left[J_{h+1}^*(s') - \underline{J}_{h+1}^m(s')\right] \\
&\geq -\sqrt{2L_m}\frac{\sqrt{\mathrm{Var}_{P(\cdot\mid s, a^*(s))}(J_{h+1}^*)} - \sqrt{\mathrm{Var}_{\bar{P}^{m-1}(\cdot\mid s, a^*(s))}(\underline{J}_{h+1}^m)}}{\sqrt{n^{m-1}(s, a^*(s)) \vee 1}} \\
&\quad + \frac{B_\star}{16H^2}\mathbb{E}_{\bar{P}^{m-1}(\cdot\mid s, a)}\left[J_{h+1}^*(s') - \underline{J}_{h+1}^m(s')\right] + \frac{13H^3 B_\star^{-1} L_m}{n^{m-1}(s, a^*(s)) \vee 1} \\
&\stackrel{(b)}{\geq} -\frac{B_\star}{16H^2}\mathbb{E}_{\bar{P}^{m-1}(\cdot\mid s, a)}\left[J_{h+1}^*(s') - \underline{J}_{h+1}^m(s')\right] - \frac{13H^2 L_m}{n^{m-1}(s, a^*(s)) \vee 1} \\
&\quad + \frac{B_\star}{16H^2}\mathbb{E}_{\bar{P}^{m-1}(\cdot\mid s, a)}\left[J_{h+1}^*(s') - \underline{J}_{h+1}^m(s')\right] + \frac{13H^3 B_\star^{-1} L_m}{n^{m-1}(s, a) \vee 1} \geq 0,
\end{aligned}
$$

where $(a)$ holds by plugging the definition of the bonuses $b_{pv1,h}^m$ and $b_p^m$ (recall Eq. (2)), as $|S| \geq 1$ by assumption, and by the induction hypothesis ($\underline{J}_{h+1}^m(s) \geq J_{h+1}^*(s)$). $(b)$ holds by Lemma B.11 while setting $\alpha = 16H^2 B_\star^{-1}$ and bounding $(5 + \alpha/2)B_\star \leq 13H^2$. Combining all the above we conclude the proof of the rightmost inequality since $J_h^*(s) - \underline{J}_h^m(s) \geq (3) \geq (4) \geq 0$.

**Rightmost inequality, pessimism.** The following relations hold.

$$
J_h^{\pi^m}(s) - \bar{J}_h^m(s) = Q_h^{\pi^m}(s, \pi_h^m(s)) - \min\{\bar{Q}_h^m(s, \pi_h^m(s)), H\}.
\tag{5}
$$

Assume that $\bar{Q}_h^m(s, \pi_h^m(s)) < H$ (otherwise, the claim holds). Then,

$$
\begin{aligned}
(5) &= Q_h^{\pi^m}(s, \pi_h^m(s)) - \bar{Q}_h^m(s, \pi_h^m(s)) \\
&= c(s, \pi_h^m(s)) - \bar{c}^{m-1}(s, \pi_h^m(s)) - b_c^m(s, \pi_h^m(s)) - b_p^m(s, \pi_h^m(s)) \\
&\quad + (P - \bar{P}^{m-1})(\cdot \mid s, \pi_h^m(s)) \cdot J_{h+1}^{\pi^m} + \mathbb{E}_{\bar{P}^{m-1}(\cdot\mid s, \pi_h^m(s))}[\underbrace{J_{h+1}^{\pi^m}(s') - \bar{J}_{h+1}^m(s')}_{\leq 0 \text{ Induction hypothesis}}] \\
&\leq -b_p^m(s, \pi_h^m(s)) + (P - \bar{P}^{m-1})(\cdot \mid s, \pi_h^m(s)) \cdot J_{h+1}^{\pi^m}.
\end{aligned}
\tag{6}
$$

We now focus on the last term. Observe that

$$(P - \bar{P}^{m-1})(\cdot \mid s, \pi_h^m(s)) \cdot J_{h+1}^{\pi^m} = (P - \bar{P}^{m-1})(\cdot \mid s, \pi_h^m(s)) \cdot J_{h+1}^* + (P - \bar{P}^{m-1})(\cdot \mid s, \pi_h^m(s)) \cdot (J_{h+1}^{\pi^m} - J_{h+1}^*)$$

$$\leq b_{pv1,h}^m(s, \pi_h^m(s)) + (P - \bar{P}^{m-1})(\cdot \mid s, \pi_h^m(s)) \cdot (J_{h+1}^{\pi^m} - J_{h+1}^*) \quad (\cap_m E^{pv1}(m) \text{ holds})$$

$$\overset{(a)}{\leq} b_{pv1,h}^m(s, \pi_h^m(s)) + \frac{36 H^3 B_\star^{-1} |S| L_m}{n^{m-1}(s, \pi_h^m(s)) \vee 1} + \frac{B_\star}{32 H^2} \mathbb{E}_{\bar{P}^{m-1}(\cdot |s, \pi_h^m(s))} \left[ (J_{h+1}^{\pi^m} - J_{h+1}^*)(s') \right]$$

$$\overset{(b)}{\leq} b_{pv1,h}^m(s, \pi_h^m(s)) + \frac{36 H^3 B_\star^{-1} |S| L_m}{n^{m-1}(s, \pi_h^m(s)) \vee 1} + \frac{B_\star}{32 H^2} \mathbb{E}_{\bar{P}^{m-1}(\cdot |s, \pi_h^m(s))} \left[ (\bar{J}_{h+1}^m - \underline{J}_{h+1}^m)(s') \right]$$

$$\overset{(c)}{\leq} \sqrt{\frac{2 \mathrm{Var}_{P(\cdot |s, \pi_h^m(s))}(J_{h+1}^*) L_m}{n^{m-1}(s, \pi_h^m(s)) \vee 1}} + \frac{41 H^3 B_\star^{-1} |S| L_m}{n^{m-1}(s, \pi_h^m(s)) \vee 1}$$

$$+ \frac{B_\star}{32 H^2} \mathbb{E}_{\bar{P}^{m-1}(\cdot |s, \pi_h^m(s))} \left[ (\bar{J}_{t-1, h+1} - \underline{J}_{h+1}^m)(s') \right],$$

where $(a)$ holds by applying Lemma B.13 while setting $\alpha = 32 H^2 B_\star^{-1}$, $C_1 = 2$, $C_2 = 2$ and bounding $2 C_2 + \alpha |S| C_1 / 2 \leq 36 H^2 B_\star^{-1} |S|$ (assumption holds since $\cap_m E^p(m)$ holds), $(b)$ holds by the induction hypothesis, and $(c)$ holds by plugging in $b_{pv1,h}^m$. Plugging this back into (6) and plugging the explicit form of the bonus $b_p^m(s, a)$ we get

$$(6) \leq -\sqrt{2 L_m} \frac{\sqrt{\mathrm{Var}_{\bar{P}^{m-1}(\cdot |s, \pi_h^m(s))}(\underline{J}_{h+1}^m)} - \sqrt{\mathrm{Var}_{P(\cdot |s, \pi_h^m(s))}(J_{h+1}^*)}}{\sqrt{n^{m-1}(s, \pi_h^m(s)) \vee 1}}$$

$$- \frac{21 H^3 B_\star^{-1} |S| L_m}{n^{m-1}(s, \pi_h^m(s)) \vee 1} - \frac{B_\star}{32 H^2} \mathbb{E}_{\bar{P}^{m-1}(\cdot |s, \pi_h^m(s))} \left[ \bar{J}_{h+1}^m(s') - \underline{J}_{h+1}^m(s') \right]$$

$$\leq \frac{B_\star}{32 H^2} \mathbb{E}_{\bar{P}^{m-1}(\cdot |s, \pi_h^m(s))} \left[ J_{h+1}^*(s') - \underline{J}_{h+1}^m(s') \right] + \frac{21 H^3 B_\star^{-1} L_m}{n^{m-1}(s, \pi_h^m(s))}$$

$$- \frac{B_\star}{32 H^2} \mathbb{E}_{\bar{P}^{m-1}(\cdot |s, \pi_h^m(s))} \left[ \bar{J}_{h+1}^m(s') - \underline{J}_{h+1}^m(s') \right] - \frac{21 H^3 B_\star^{-1} |S| L_m}{n^{m-1}(s, \pi_h^m(s))} = 0,$$

where the last inequality holds by Lemma B.11 while setting $\alpha = 32 H^2 B_\star^{-1}$ and bounding $(5 + \alpha/2) B_\star \leq 21 H^3 B_\star^{-1}$. Combining all the above we concludes the proof as

$$J_h^{\pi^m}(s) - \bar{J}_h^m(s) \leq (5) \leq (6) \leq 0. \qquad \square$$

Finally, using similar techniques to Efroni et al. [2021], we can prove an additional high probability bounds which hold alongside the basic good event $\mathbb{G}_1$.

**Lemma B.3** (The Good Event). *Let $\mathbb{G}_1$ be the event defined in Lemma B.1, and define the following random variables.*

$$Y_{1,h}^m = \bar{J}_h^m(s_h^m) - \underline{J}_h^m(s_h^m)$$

$$Y_{2,h}^m = \mathrm{Var}_{P(\cdot |s_h^m, a_h^m)}(J_{h+1}^{\pi^m})$$

$$Y_3^m = \left( \sum_{h=1}^{H} c(s_h^m, a_h^m) + c_f(s_{h+1}^m) \right)^2$$

$$Y_4^m = \left( \sum_{h=1}^{h_m} c(s_h^m, a_h^m) + c_f(s_{h+1}^m) \mathbb{I}\{h_m = H\} \right)^2$$

$$Y_5^m = \sum_{h=1}^{h_m} c(s_h^m, a_h^m) + c_f(s_{h+1}^m) \mathbb{I}\{h_m = H\}.$$

The second good event is the intersection of two events $\mathbb{G}_2 = E^{OP} \cap E^{\text{Var}} \cap E^{Sec1} \cap E^{Sec2} \cap E^{cost}$ defined as follows.

$$E^{OP} = \left\{ \forall h \in [H], M \geq 1 : \sum_{m=1}^{M} \mathbb{E}[Y_{1,h}^m \mid \bar{U}_h^m] \leq 68H^2 L_M + \left(1 + \frac{1}{4H}\right) \sum_{m=1}^{M} Y_{1,h}^m \right\}$$

$$E^{\text{Var}} = \left\{ \forall M \geq 1 : \sum_{m=1}^{M} \sum_{h=1}^{H} Y_{2,h}^m \leq 16H^3 L_M + 2 \sum_{m=1}^{M} \sum_{h=1}^{H} \mathbb{E}[Y_{2,h}^m \mid \bar{U}^m] \right\}$$

$$E^{Sec1} = \left\{ \forall M \geq 1 : \sum_{m=1}^{M} \mathbb{E}[Y_3^m \mid \bar{U}^m] \leq 68H^4 L_M + 2 \sum_{m=1}^{M} Y_3^m \right\}$$

$$E^{Sec2} = \left\{ \forall M \geq 1 : \sum_{m=1}^{M} Y_4^m \leq 16H^4 L_M + 2 \sum_{m=1}^{M} \mathbb{E}[Y_4^m \mid \bar{U}^m] \right\}$$

$$E^{cost} = \left\{ \forall M \geq 1 : \sum_{m=1}^{M} Y_5^m \leq 8H L_M + 2 \sum_{m=1}^{M} \mathbb{E}[Y_5^m \mid \bar{U}^m] \right\}.$$

Then, the good event $\mathbb{G} = \mathbb{G}_1 \cap \mathbb{G}_2$ holds with probability at least $1 - \delta$.

*Proof.* **Event $E^{OP}$.** Fix $h$ and $M$. We start by defining the random variable $W^m = \mathbb{I}\{\bar{J}_h^m(s) - \underline{J}_h^m(s) \geq 0 \; \forall h \in [H], s \in S\}$. Observe that $Y_h^m$ is $\bar{U}_h^m$ measurable and also notice that $W^m$ is $\bar{U}^m$ measurable, as both $\pi^m$ and $\bar{J}_h^m$ are $\bar{U}^m$-measurable. Finally, define $\tilde{Y}^m = W^m Y_h^m$. Importantly, notice that $\tilde{Y}^m \in [0, 2H]$ almost surely, by definition of $W^m$ and since $\bar{J}_h^m(s), \underline{J}_h^m(s) \in [0, 2H]$ by the update rule. Thus, using Lemma E.2 with $C = 2H \geq 1$, we get

$$\sum_{m=1}^{M} \mathbb{E}[\tilde{Y}_h^m \mid \bar{U}_h^m] \leq \left(1 + \frac{1}{4H}\right) \sum_{m=1}^{M} \tilde{Y}_h^m + 68H^2 \log \frac{2HM(M+1)}{\delta},$$

with probability greater than $1 - \delta$, and since $W^m$ is $\bar{U}^m$-measurable, we can write

$$\sum_{m=1}^{M} W^m \mathbb{E}[Y_h^m \mid \bar{U}_h^m] \leq \left(1 + \frac{1}{4H}\right) \sum_{m=1}^{M} W^m Y_h^m + 68H^2 \log \frac{2HM(M+1)}{\delta}. \tag{7}$$

Importantly, notice that under $\mathbb{G}_1$, it holds that $W^m \equiv 1$ (by Lemma B.2). Therefore, applying the union bound and setting $\delta = \delta/(2HM(M+1))$ we get

$$\mathbb{P}(\overline{E^O} \cap \mathbb{G}_1) \leq$$

$$\leq \sum_{h=1}^{H} \sum_{M=1}^{\infty} \mathbb{P}\left( \left\{ \sum_{m=1}^{M} \mathbb{E}[Y_h^m \mid \bar{U}_h^m] \geq \left(1 + \frac{1}{4H}\right) \sum_{m=1}^{M} Y_h^m + 68H^2 \log \frac{2HM(M+1)}{\delta} \right\} \cap \mathbb{G}_1 \right)$$

$$= \sum_{h=1}^{H} \sum_{M=1}^{\infty} \mathbb{P}\left( \left\{ \sum_{m=1}^{M} W^m \mathbb{E}[Y_h^m \mid \bar{U}_h^m] \geq \left(1 + \frac{1}{4H}\right) \sum_{m=1}^{M} W^m Y_h^m + 68H^2 \log \frac{2HM(M+1)}{\delta} \right\} \cap \mathbb{G}_1 \right)$$

$$\leq \sum_{h=1}^{H} \sum_{M=1}^{\infty} \mathbb{P}\left( \sum_{m=1}^{M} W^m \mathbb{E}[Y_h^m \mid \bar{U}_h^m] \geq \left(1 + \frac{1}{4H}\right) \sum_{m=1}^{M} W^m Y_h^m + 68H^2 \log \frac{2HM(M+1)}{\delta} \right)$$

$$\leq \sum_{h=1}^{H} \sum_{M=1}^{\infty} \frac{\delta}{2HM(M+1)} = \delta/2,$$

where the first relation is by a union bound, the second relation follows because $W^m \equiv 1$ under $\mathbb{G}_1$, and the last relation is by (7). Finally, we have

$$\mathbb{P}(\overline{\mathbb{G}}) \leq \mathbb{P}(\overline{\mathbb{G}_2} \cap \mathbb{G}_1) + 2\mathbb{P}(\overline{\mathbb{G}_1}) \leq \frac{\delta}{2} + \frac{2\delta}{4} = \delta.$$

Replacing $\delta \to \delta/5$ implies that $\mathbb{P}(\overline{E^{OP}} \cap \mathbb{G}_1) \leq \frac{\delta}{10}$.

**Event $E^{\text{Var}}$.** Fix $h \in [H]$. Observe that $Y_{2,h}^m$ is $\bar{U}^m$ measurable and that $0 \le Y_{2,h}^m \le 4H^2$. Applying the second statement of Lemma E.2 we get that

$$\sum_{m=1}^{M} Y_{2,h}^m \le 2 \sum_{m=1}^{M} \mathbb{E}[Y_{2,h}^m | \bar{U}^m] + 16H^2 \log \frac{1}{\delta}.$$

By taking union bound, as in the proof of the first statement of the lemma on all $h \in [H]$ and summing over $h \in [H]$, we get that with probability at least $1 - \delta/10$ for all $M \ge 1$ it holds that

$$\sum_{m=1}^{M} \sum_{h=1}^{H} Y_{2,h}^m \le 2 \sum_{m=1}^{M} \sum_{h=1}^{H} \mathbb{E}[Y_{2,h}^m | \bar{U}^m] + 16H^3 L_M.$$

**Event $E^{Sec1}$.** Observe that $Y_3^m$ is $\bar{U}^m$ measurable and that $0 \le Y_3^m \le 4H^2$. Applying the first statement of Lemma E.2 we get that

$$\sum_{m=1}^{M} \mathbb{E}[Y_3^m | \bar{U}^m] \le 2 \sum_{m=1}^{M} Y_3^m + 50H^4 \log \frac{1}{\delta}.$$

By taking union bound we get that with probability at least $1 - \delta/10$ the event holds.

**Event $E^{Sec2}$.** Observe that $Y_4^m$ is $\bar{U}^m$ measurable and that $0 \le Y_4^m \le 4H^2$. Applying the second statement of Lemma E.2 we get that

$$\sum_{m=1}^{M} Y_4^m \le 2 \sum_{m=1}^{M} \mathbb{E}[Y_4^m | \bar{U}^m] + 16H^2 \log \frac{1}{\delta}.$$

By taking union bound we get that with probability at least $1 - \delta/10$ the event holds.

**Event $E^{cost}$.** Observe that $Y_5^m$ is $\bar{U}^m$ measurable and that $0 \le Y_5^m \le 2H$. Applying the second statement of Lemma E.2 we get that

$$\sum_{m=1}^{M} Y_5^m \le 2 \sum_{m=1}^{M} \mathbb{E}[Y_5^m | \bar{U}^m] + 8H \log \frac{1}{\delta}.$$

By taking union bound we get that with probability at least $1 - \delta/10$ the event holds.

**Combining all the above.** We bound the probability of $\overline{G}$ as follows:

$$\mathbb{P}(\overline{\mathbb{G}}) \le \mathbb{P}(\overline{\mathbb{G}_1}) + \mathbb{P}(\overline{E^{OP}} \cap \mathbb{G}_1) + \mathbb{P}(\overline{E^{\text{Var}}}) + \mathbb{P}(\overline{E^{Sec1}}) + \mathbb{P}(\overline{E^{Sec2}}) + \mathbb{P}(\overline{E^{cost}}) \le \frac{\delta}{2} + 5 \cdot \frac{\delta}{10} = \delta.$$

$\square$

## B.2  ULCVI is admissible

By the definition of the algorithm and its regret bound in Theorem 5.1, it is clear that properties 1,2,3 of the admissible algorithm definition hold. Thus, it remains to show property 4 by bounding $\omega_{\text{ULCVI}}$. In order to show that $\omega_{\text{ULCVI}} = O(H^4 B_\star^{-2} |S|)$, we need to show that if the number of visits to $(s, a)$ is at least $\alpha H^4 B_\star^{-2} |S| \log \frac{MH|S||A|}{\delta}$ (for a large enough universal constant $\alpha > 0$) then $\|P(\cdot \mid s, a) - \widetilde{P}_t(\cdot \mid s, a)\|_1 \le 1/(18H)$ and $|c(s, a) - \tilde{c}_h^t(s, a)| \le B_\star/H$ (under the good event), where $\widetilde{P}, \tilde{c}$ are the estimations used by the algorithm to compute its optimistic $Q$-function (i.e., these are the empirical transition estimate and the empirical cost estimate plus the bonus).

Indeed, by event $\cap_{m>0} E^p(m)$,

$$\|P(\cdot \mid s, a) - \widetilde{P}(\cdot \mid s, a)\|_1 = \|P(\cdot \mid s, a) - \bar{P}(\cdot \mid s, a)\|_1$$

$$\le \sqrt{\frac{2|S| \log \frac{16M^3 H|S|^2|A|}{\delta}}{n(s, a)}} + \frac{2|S| \log \frac{16M^3 H|S|^2|A|}{\delta}}{n(s, a)}$$

$$\le \frac{4B_\star}{\sqrt{\alpha}H^2} + \frac{16B_\star^2}{\alpha H^4} \le \frac{1}{18H},$$

for $\alpha > 5800$, where the first inequality holds by Jensen inequality and since event $\cap_{m>0} E^p(m)$ holds. By the definition of the exploration bonuses we have

$$|c(s,a) - \tilde{c}_h(s,a)| \leq |c(s,a) - \bar{c}(s,a)| + b_c(s,a) + b_p(s,a)$$

$$\leq 3\sqrt{\frac{2B_\star^2 \log \frac{16M^3 H|S|^2|A|}{\delta}}{n(s,a)}} + \frac{72H^3 B_\star^{-1}|S| \log \frac{16M^3 H|S|^2|A|}{\delta}}{n(s,a)} + \frac{B_\star \max_{s'} \bar{J}_{h+1}(s') - \underline{J}_{h+1}(s')}{16H^2}$$

$$\leq \frac{12B_\star^2}{\sqrt{\alpha}H^2} + \frac{800B_\star}{\alpha H} + \frac{B_\star}{16H} \leq \frac{B_\star}{H},$$

for $\alpha > 5800$.

Finally, note that although our algorithm does not update the policy in the beginning of every episode (only when the number of visits to some state-action pair is doubled), this only implies that the constant $\alpha$ needs to be doubled.

## B.3   Proof of Theorem 5.1

As in the proof of UCBVI, before establishing the proof of Theorem 5.1 we establish the following key lemma that bounds the on-policy errors at time step $h$ by the on-policy errors at time step $h+1$ and additional additive terms. Given this result, the analysis follows with relative ease.

**Lemma B.4** (ULCBVI, Key Recursion Bound). *Conditioning on the good event $\mathbb{G}$, the following bound holds for all $h \in [H]$.*

$$\sum_{m=1}^{M} \bar{J}_h^m(s_h^m) - \underline{J}_h^m(s_h^m) \leq 68H^2 L_M + \sum_{m=1}^{M} \frac{310H^3 B_\star^{-1}|S|L_m}{n^{m-1}(s_h^m, a_h^m) \vee 1} + \sum_{m=1}^{M} 4\sqrt{L_m} \frac{\sqrt{c(s_h^m, a_h^m)}}{\sqrt{n^{m-1}(s_h^m, a_h^m) \vee 1}}$$

$$+ \sum_{m=1}^{M} 2\sqrt{2L_m} \frac{\sqrt{\mathrm{Var}_{P(\cdot|s_h^m, a_h^m)}(J_{h+1}^{\pi^m})}}{\sqrt{n^{m-1}(s_h^m, a_h^m) \vee 1}} + \left(1 + \frac{1}{2H}\right)^2 \sum_{m=1}^{M} \left(\bar{J}_{h+1}^m(s_{h+1}^m) - \underline{J}_{h+1}^m(s_{h+1}^m)\right).$$

*Proof.* We bound each of the terms in the sum as follows.

$$\bar{J}_h^m(s_h^m) - \underline{J}_h^m(s_h^m) = 2b_c^m(s_h^m, a_h^m) + 2b_p^m(s_h^m, a_h^m) + \mathbb{E}_{\bar{P}^{m-1}(\cdot|s_h^m, a_h^m)}[\bar{J}_{h+1}^m(s_{h+1}^m) - \underline{J}_{h+1}^m(s_{h+1}^m)]$$

$$= 2b_c^m(s_h^m, a_h^m) + 2b_p^m(s_h^m, a_h^m)$$

$$+ \mathbb{E}_{P(\cdot|s_h^m, a_h^m)}[\bar{J}_{h+1}^m(s_{h+1}^m) - \underline{J}_{h+1}^m(s_{h+1}^m)] + (\bar{P}^{m-1} - P)(\cdot|s_h^m, a_h^m) \cdot \left(\bar{J}_{h+1}^m - \underline{J}_{h+1}^m\right)$$

$$\leq 2b_c^m(s_h^m, a_h^m) + 2b_p^m(s_h^m, a_h^m)$$

$$+ \frac{8H^2|S|L_m}{n^{m-1}(s_h^m, a_h^m) \vee 1} + \left(1 + \frac{1}{4H}\right) \mathbb{E}_{P(\cdot|s_h^m, a_h^m)}[\bar{J}_{h+1}^m(s_{h+1}^m) - \underline{J}_{h+1}^m(s_{h+1}^m)], \qquad (8)$$

where the last relation holds by Lemma B.13 which upper bounds

$$(\bar{P}^{m-1} - P)(\cdot|s_h^m, a_h^m) \cdot \left(\bar{J}_{h+1}^m - \underline{J}_{h+1}^m\right) \leq \frac{8H^2|S|L_m}{n^{m-1}(s_h^m, a_h^m) \vee 1} + \frac{1}{4H} \mathbb{E}_{P(\cdot|s_h^m, a_h^m)}[\bar{J}_{h+1}^m(s_{h+1}^m) - \underline{J}_{h+1}^m(s_{h+1}^m)]$$

by setting $\alpha = 4H, C_1 = C_2 = 2$ and bounding $HL_m(2C_2 + \alpha|S|C_1/2) \leq 8H^2|S|L_m$ (the assumption of the lemma holds since the event $\cap_m E^p(m)$ holds). Taking the sum over $m \in [M]$ we get that

$$\sum_{m=1}^{M} \bar{J}_h^m(s_h^m) - \underline{J}_h^m(s_h^m) \leq \sum_{m=1}^{M} 2b_c^m(s_h^m, a_h^m) + \sum_{m=1}^{M} 2b_p^m(s_h^m, a_h^m)$$

$$+ \sum_{m=1}^{M} \frac{8H^2|S|L_m}{n^{m-1}(s_h^m, a_h^m) \vee 1} + \sum_{m=1}^{M} \left(1 + \frac{1}{4H}\right) \mathbb{E}_{P(\cdot|s_h^m, a_h^m)}[\bar{J}_{h+1}^m(s_{h+1}^m) - \underline{J}_{h+1}^m(s_{h+1}^m)].$$

$$(9)$$

The first sum is bounded in Lemma B.5 by

$$\sum_{m=1}^{M} b_c^m(s_h^m, a_h^m) \leq \sum_{m=1}^{M} \sqrt{\frac{2c(s_h^m, a_h^m)L_m}{n^{m-1}(s_h^m, a_h^m) \vee 1}} + \sum_{m=1}^{M} \frac{10L_m}{n^{m-1}(s_h^m, a_h^m) \vee 1},$$

and the second sum is bounded in Lemma B.6 by

$$\sum_{m=1}^{M} b_p^m(s_h^m, a_h^m) \leq \sum_{m=1}^{M} \frac{139 H^3 B_\star^{-1} |S| L_m}{n^{m-1}(s_h^m, a_h^m) \vee 1} + \sum_{m=1}^{M} \sqrt{2 L_m} \frac{\sqrt{\mathrm{Var}_{P(\cdot|s_h^m, a_h^m)}(J_{h+1}^{\pi^m})}}{\sqrt{n^{m-1}(s_h^m, a_h^m) \vee 1}}$$

$$+ \frac{1}{8H} \sum_{m=1}^{M} \mathbb{E}_{P(\cdot|s_h^m, a_h^m)}[\bar{J}_{h+1}^m(s_{h+1}^m) - \underline{J}_{h+1}^m(s_{h+1}^m)].$$

Plugging this into (9) and rearranging the terms we get

$$\sum_{m=1}^{M} \bar{J}_h^m(s_h^m) - \underline{J}_h^m(s_h^m) \leq \sum_{m=1}^{M} \frac{2\sqrt{2c(s_h^m, a_h^m) L_m}}{\sqrt{n^{m-1}(s_h^m, a_h^m) \vee 1}} + \sum_{m=1}^{M} 2\sqrt{2L_m} \frac{\sqrt{\mathrm{Var}_{P(\cdot|s_h^m, a_h^m)}(J_{h+1}^{\pi^m})}}{\sqrt{n^{m-1}(s_h^m, a_h^m) \vee 1}}$$

$$+ \sum_{m=1}^{M} \frac{286 H^3 B_\star^{-1} |S| L_m}{n^{m-1}(s_h^m, a_h^m) \vee 1} + \left(1 + \frac{1}{2H}\right) \sum_{m=1}^{M} \mathbb{E}_{P(\cdot|s_h^m, a_h^m)}[\bar{J}_{h+1}^m(s_{h+1}^m) - \underline{J}_{h+1}^m(s_{h+1}^m)]$$

$$\leq 68 H^2 L_M + \sum_{m=1}^{M} \frac{2\sqrt{2L_m}}{\sqrt{n^{m-1}(s_h^m, a_h^m) \vee 1}} + \sum_{m=1}^{M} \frac{286 H^3 B_\star^{-1} |S| L_m}{n^{m-1}(s_h^m, a_h^m) \vee 1}$$

$$+ \sum_{m=1}^{M} 2\sqrt{2L_m} \frac{\sqrt{\mathrm{Var}_{P(\cdot|s_h^m, a_h^m)}(J_{h+1}^{\pi^m})}}{\sqrt{n^{m-1}(s_h^m, a_h^m) \vee 1}} + \left(1 + \frac{1}{2H}\right)^2 \sum_{m=1}^{M} \bar{J}_{h+1}^m(s_{h+1}^m) - \underline{J}_{h+1}^m(s_{h+1}^m),$$

where the last inequality follows since the second good event holds. $\qquad\square$

*Proof of Theorem 5.1.* Start by conditioning on the good event which holds with probability greater than $1 - \delta$. Applying the optimism-pessimism of the upper and lower value function we get

$$\sum_{m=1}^{M} J_1^{\pi^m}(s_1^m) - J_1^*(s_1^m) \leq \sum_{m=1}^{M} \bar{J}_1^m(s_1^m) - \underline{J}_1^m(s_1^m). \tag{10}$$

Iteratively applying Lemma B.4 and bounding the exponential growth by $\left(1 + \frac{1}{2H}\right)^{2H} \leq e \leq 3$, the following upper bound on the cumulative regret is obtained.

$$(10) \leq 204 H^3 B_\star^{-1} L_M + \sum_{m=1}^{M} \sum_{h=1}^{H} \frac{930 H^3 B_\star^{-1} |S| L_m}{n^{m-1}(s_h^m, a_h^m) \vee 1}$$

$$+ \sum_{m=1}^{M} \sum_{h=1}^{H} \frac{12\sqrt{c(s_h^m, a_h^m) L_m}}{\sqrt{n^{m-1}(s_h^m, a_h^m) \vee 1}} + 9 \sum_{m=1}^{M} \sum_{h=1}^{H} \frac{\sqrt{L_m \mathrm{Var}_{P(\cdot|s_h^m, a_h^m)}(J_{h+1}^{\pi^m})}}{\sqrt{n^{m-1}(s_h^m, a_h^m)}}. \tag{11}$$

We now bound each of the three sums in Eq. (11). We bound the first sum in Eq. (11) via standard analysis as follows:

$$\sum_{m=1}^{M} \sum_{h=1}^{H} \frac{H^3 B_\star^{-1} |S| L_m}{n^{m-1}(s_h^m, a_h^m) \vee 1} \leq H^3 B_\star^{-1} |S| L_M \sum_{m=1}^{M} \sum_{h=1}^{H} \frac{1}{n^{m-1}(s_h^m, a_h^m) \vee 1}$$

$$= H^3 B_\star^{-1} |S| L_M \sum_{m=1}^{M} \sum_{s,a} \frac{\sum_{h=1}^{H} \mathbb{I}\{s_h^m = s, a_h^m = a\}}{n^{m-1}(s, a) \vee 1}$$

$$\leq H^3 B_\star^{-1} |S| L_M \sum_{m=1}^{M} \sum_{s,a} \mathbb{I}\{n^{m-1}(s, a) \geq H\} \frac{\sum_{h=1}^{H} \mathbb{I}\{s_h^m = s, a_h^m = a\}}{n^{m-1}(s, a) \vee 1} + 2H^4 B_\star^{-1} |S|^2 |A| L_M$$

$$\leq 3 H^3 B_\star^{-1} |S|^2 |A| L_M \log(MH) + 2H^4 B_\star^{-1} |S|^2 |A| L_M,$$

where the last inequality is by Lemma B.12 that bounds $\sum_{m,s,a} \mathbb{I}\{n^{m-1}(s, a) \geq H\} \frac{\sum_{h=1}^{H} \mathbb{I}\{s_h^m = s, a_h^m = a\}}{n^{m-1}(s,a) \vee 1} \leq 3|S||A| \log(MH)$.

The second sum in Eq. (11) is bounded as follows.

$$\sum_{m=1}^{M}\sum_{h=1}^{H}\frac{\sqrt{c(s_h^m,a_h^m)L_m}}{\sqrt{n^{m-1}(s_h^m,a_h^m)\vee 1}} \leq \sum_{m=1}^{M}\sum_{h=1}^{H}\frac{\sqrt{c(s_h^m,a_h^m)L_m}}{\sqrt{n^{m-1}(s_h^m,a_h^m)\vee 1}}\mathbb{I}\{n^{m-1}(s_h^m,a_h^m)\geq H\} + 2H|S||A|L_M$$

$$\overset{(a)}{\leq}\sqrt{L_M}\sqrt{\sum_{m=1}^{M}\sum_{h=1}^{H}c(s_h^m,a_h^m)}\cdot\sqrt{\sum_{m=1}^{M}\sum_{h=1}^{H}\frac{\mathbb{I}\{n^{m-1}(s_h^m,a_h^m)\geq H\}}{n^{m-1}(s_h^m,a_h^m)\vee 1}} + 2H|S||A|L_M$$

$$\overset{(b)}{\leq}\sqrt{L_M}\sqrt{\sum_{m=1}^{M}\sum_{h=1}^{H}c(s_h^m,a_h^m)}\cdot\sqrt{3|S||A|\log(MH)} + 2H|S||A|L_M$$

$$\leq\sqrt{3|S||A|}L_M\sqrt{\sum_{m=1}^{M}\sum_{h=1}^{H}c(s_h^m,a_h^m) + c_f(s_{H+1}^m)} + 2H|S||A|L_M$$

$$\leq O\left(\sqrt{B_\star|S||A|M}L_M + H^3 B_\star^{-1}|S|^2|A|\log^{3/2}\frac{MH|S||A|}{\delta}\right).$$

where (a) is by Cauchy-Schwartz, (b) is by Lemma B.12, and the last inequality is by Lemma B.7. The third sum in Eq. (11) is bounded in Lemma B.8 by

$$\sum_{m=1}^{M}\sum_{h=1}^{H}\frac{\sqrt{L_m\text{Var}_{P(\cdot|s_h^m,a_h^m)}(J_{h+1}^{\pi^m})}}{\sqrt{n^{m-1}(s_h^m,a_h^m)}} \leq \sqrt{L_M}\sum_{m=1}^{M}\sum_{h=1}^{H}\frac{\sqrt{\text{Var}_{P(\cdot|s_h^m,a_h^m)}(J_{h+1}^{\pi^m})}}{\sqrt{n^{m-1}(s_h^m,a_h^m)}} \qquad (L_m \text{ increasing in } m)$$

$$\leq\sqrt{L_m}\cdot O\left(\sqrt{B_\star^2|S||A|M\log(MH)} + H^3 B_\star^{-1}|S|^2|A|\log\frac{MH|S||A|}{\delta}\right). \qquad (\text{Lemma B.8})$$

$\square$

## B.4 Bounds on the cumulative bonuses

**Lemma B.5** (Bound on the Cumulative Cost Function Bonus). *Conditioning on the good event the following bound holds for all $h \in [H]$.*

$$\sum_{m=1}^{M}b_c^m(s_h^m,a_h^m) \leq \sum_{m=1}^{M}\sqrt{\frac{2c(s_h^m,a_h^m)L_m}{n^{m-1}(s_h^m,a_h^m)\vee 1}} + \sum_{m=1}^{M}\frac{10L_m}{n^{m-1}(s_h^m,a_h^m)\vee 1}.$$

*Proof.* By definition of $b_c^m$ and since the event $\cap_m E^{cv}(m)$ holds, we have

$$\sum_{m=1}^{M}b_c^m(s_h^m,a_h^m) = \sum_{m=1}^{M}\sqrt{\frac{2\overline{\text{Var}}_{s_h^m,a_h^m}^{m-1}(c)L_m}{n^{m-1}(s_h^m,a_h^m)\vee 1}} + \frac{5L_m}{n^{m-1}(s_h^m,a_h^m)\vee 1}$$

$$\leq\sum_{m=1}^{M}\sqrt{\frac{2\text{Var}_{s_h^m,a_h^m}(c)L_m}{n^{m-1}(s_h^m,a_h^m)\vee 1}} + \sqrt{\frac{2L_m\,|\text{Var}_{s_h^m,a_h^m}(c) - \overline{\text{Var}}_{s_h^m,a_h^m,t-1}^{m-1}(c)|}{n^{m-1}(s_h^m,a_h^m)\vee 1}} + \frac{5L_m}{n^{m-1}(s_h^m,a_h^m)\vee 1}$$

$$\leq\sum_{m=1}^{M}\sqrt{\frac{2\text{Var}_{s_h^m,a_h^m}(c)L_m}{n^{m-1}(s_h^m,a_h^m)\vee 1}} + \frac{10L_m}{n^{m-1}(s_h^m,a_h^m)\vee 1},$$

where the first inequality holds since $\sqrt{a+b}\leq\sqrt{|a|}+\sqrt{|b|}$. Finally, notice that for every $(s,a)\in S\times A$ the variance of the cost is bounded by the second moment, which is bounded by the expected value $c(s,a)$ since the random cost value is bounded in $[0,1]$. $\square$

**Lemma B.6** (Bound on the Cumulative Transition Model Bonus). *Conditioning on the good event the following bound holds for all $h \in [H]$.*

$$\sum_{m=1}^{M} b_p^m(s_h^m, a_h^m) \leq \sum_{m=1}^{M} \frac{139 H^3 B_\star^{-1} |S| L_m}{n^{m-1}(s_h^m, a_h^m) \vee 1} + \sum_{m=1}^{M} \sqrt{2 L_m} \frac{\sqrt{\mathrm{Var}_{P(\cdot|s_h^m, a_h^m)}(J_{h+1}^{\pi^m})}}{\sqrt{n^{m-1}(s_h^m, a_h^m) \vee 1}}$$

$$+ \frac{1}{8H} \sum_{m=1}^{M} \mathbb{E}_{P(\cdot|s_h^m, a_h^m)}[\bar{J}_{h+1}^m(s_{h+1}^m) - \underline{J}_{h+1}^m(s_{h+1}^m)].$$

*Proof.* First, by applying Lemma B.13 with $\alpha = 8H, C_1 = C_2 = 2$ and $H L_m(2 C_2 + \alpha |S| C_1/2) \leq 12 H^2 |S| L_m$, we have

$$\mathbb{E}_{\bar{P}^{m-1}(\cdot|s,a)}[\bar{J}_{h+1}^m(s') - \underline{J}_{h+1}^m(s')] = \mathbb{E}_{P(\cdot|s,a)}[\bar{J}_{h+1}^m(s') - \underline{J}_{h+1}^m(s')] + (\bar{P}^{m-1} - P)(\cdot \mid s, a) \cdot (\bar{J}_{h+1}^m - \underline{J}_{h+1}^m)$$

$$\leq \frac{9}{8} \mathbb{E}_{P(\cdot|s,a)}[\bar{J}_{h+1}^m(s') - \underline{J}_{h+1}^m(s')] + \frac{12 H^2 |S| L_m}{n^{m-1}(s,a) \vee 1}. \tag{12}$$

Thus, the bonus $b_t^p(s, a)$ can be upper bounded as follows.

$$b_p^m(s, a) \leq \sqrt{2} \sqrt{\frac{\mathrm{Var}_{\bar{P}^{m-1}(\cdot|s,a)}(\underline{J}_{h+1}^m) L_m}{n^{m-1}(s,a) \vee 1}} + \frac{1}{16H} \mathbb{E}_{\bar{P}^{m-1}(\cdot|s,a)}[\bar{J}_{h+1}^m(s') - \underline{J}_{h+1}^m(s')] + \frac{62 H^3 B_\star^{-1} |S| L_m}{n^{m-1}(s,a) \vee 1}$$

$$\leq \sqrt{2} \sqrt{\frac{\mathrm{Var}_{\bar{P}^{m-1}(\cdot|s,a)}(\underline{J}_{h+1}^m) L_m}{n^{m-1}(s,a) \vee 1}} + \frac{9}{128H} \mathbb{E}_{P(\cdot|s,a)}[\bar{J}_{h+1}^m(s') - \underline{J}_{h+1}^m(s')] + \frac{74 H^3 B_\star^{-1} |S| L_m}{n^{m-1}(s,a) \vee 1}. \tag{13}$$

We bound the first term of (13) to establish the lemma. It holds that

$$\sqrt{2 L_m} \sqrt{\frac{\mathrm{Var}_{\bar{P}^{m-1}(\cdot|s,a)}(\underline{J}_{h+1}^m)}{n^{m-1}(s,a) \vee 1}} = \underbrace{\sqrt{2 L_m} \frac{\sqrt{\mathrm{Var}_{\bar{P}^{m-1}(\cdot|s,a)}(\underline{J}_{h+1}^m)} - \sqrt{\mathrm{Var}_{P(\cdot|s,a)}(J_{h+1}^*)}}{\sqrt{n^{m-1}(s,a) \vee 1}}}_{(i)}$$

$$+ \underbrace{\sqrt{2 L_m} \frac{\sqrt{\mathrm{Var}_{P(\cdot|s,a)}(J_{h+1}^*)} - \sqrt{\mathrm{Var}_{P(\cdot|s,a)}(J_{h+1}^{\pi^m})}}{\sqrt{n^{m-1}(s,a) \vee 1}}}_{(ii)}$$

$$+ \frac{\sqrt{2 L_m} \sqrt{\mathrm{Var}_{P(\cdot|s,a)}(J_{h+1}^{\pi^m})}}{\sqrt{n^{m-1}(s,a) \vee 1}}.$$

Term $(i)$ is bounded by Lemma B.11 (by setting $\alpha = 32H$ and $(5 + \alpha/2) B_\star \leq 21 H^2$),

$$\sqrt{2 L_m} \frac{\sqrt{\mathrm{Var}_{\bar{P}^{m-1}(\cdot|s,a)}(\underline{J}_{h+1}^m)} - \sqrt{\mathrm{Var}_{P(\cdot|s,a)}(J_{h+1}^*)}}{\sqrt{n^{m-1}(s,a) \vee 1}} \leq \frac{1}{32H} \mathbb{E}_{\bar{P}^{m-1}(\cdot|s,a)}[J_{h+1}^*(s') - \underline{J}_{h+1}^m(s')] + \frac{21 H^2 L_m}{n^{m-1}(s,a) \vee 1}.$$

Following the same steps as in (12), we get

$$\mathbb{E}_{\bar{P}^{m-1}(\cdot|s,a)}[J_{h+1}^*(s') - \underline{J}_{h+1}^m(s')] \leq \frac{9}{8} \mathbb{E}_{P(\cdot|s,a)}[J_{h+1}^*(s') - \underline{J}_{h+1}^m(s')] + \frac{12 H^2 |S| L_m}{n^{m-1}(s,a) \vee 1},$$

and thus,

$$(i) \leq \frac{9}{256H} \mathbb{E}_{P(\cdot|s,a)}[J_{h+1}^*(s') - \underline{J}_{h+1}^m(s')] + \frac{33 H^2 |S| L_m}{n^{m-1}(s,a) \vee 1}.$$

Term $(ii)$ is bounded as follows.

$$(ii) \leq \frac{\sqrt{\mathrm{Var}_{P(\cdot|s,a)}(J_{h+1}^* - J_{h+1}^{\pi^m})}}{\sqrt{n^{m-1}(s,a) \vee 1}} \qquad \text{(By Lemma E.3)}$$

$$\leq \frac{\sqrt{\mathbb{E}_{P(\cdot|s,a)}[(J_{h+1}^*(s') - J_{h+1}^{\pi^m}(s'))^2]}}{\sqrt{n^{m-1}(s,a) \vee 1}}$$

$$\leq \frac{\sqrt{2H \mathbb{E}_{P(\cdot|s,a)}[(J_{h+1}^*(s') - J_{h+1}^{\pi^m}(s'))]}}{\sqrt{n^{m-1}(s,a) \vee 1}} \qquad (0 \leq J_h^*(s') - V_h^{\pi^m}(s') \leq 2H)$$

$$\leq \frac{1}{64H} \mathbb{E}_{P(\cdot|s,a)}[(J_{h+1}^{\pi^m}(s') - J_{h+1}^*(s'))] + \frac{32 H^2}{n^{m-1}(s,a) \vee 1}. \qquad (ab \leq \frac{1}{\alpha} a^2 + \frac{\alpha}{4} b^2 \text{ for } \alpha = 64H)$$

Thus, applying $\bar{J}_h^m \geq J_h^{\pi^m} \geq J_h^* \geq \underline{J}_h^m$ (Lemma B.2) in the bounds of (i) and (ii) we get

$$b_p^m(s,a) \leq \frac{1}{8H}\mathbb{E}_{P(\cdot|s,a)}[(\bar{J}_h^m(s') - \underline{J}_h^m(s'))] + \frac{139H^3 B_\star^{-1}|S|L_m}{n^{m-1}(s,a)\vee 1} + \frac{\sqrt{2L_m}\sqrt{\mathrm{Var}_{P(\cdot|s,a)}(J_{h+1}^{\pi^m})}}{\sqrt{n^{m-1}(s,a)\vee 1}},$$

and summing over $m$ concludes the proof. $\square$

**Lemma B.7** (Bound on Cost Term). *Conditioning on the good event, it holds that*

$$\sum_{m=1}^{M}\sum_{h=1}^{H} c(s_h^m, a_h^m) + c_f(s_{H+1}^m) \leq O\left(B_\star M + H^5 B_\star^{-2}|S|^2|A|\log\frac{MH|S||A|}{\delta}\right).$$

*Proof.* Denote by $h_m$ the last time step before reaching an unknown state-action pair (or $H$ if it was not reached). By the event $E^{cost}$ we have

$$\sum_{m=1}^{M}\sum_{h=1}^{H} c(s_h^m, a_h^m) + c_f(s_{H+1}^m) = \sum_{m=1}^{M}\left(\sum_{h=h_m+1}^{H} c(s_h^m, a_h^m) + c_f(s_{h+1}^m)\mathbb{I}\{h_m \neq H\}\right)$$

$$+ \sum_{m=1}^{M}\left(\sum_{h=1}^{h_m} c(s_h^m, a_h^m) + c_f(s_{h+1}^m)\mathbb{I}\{h_m = H\}\right)$$

$$\leq 2\alpha H^5 B_\star^{-2}|S|^2|A|\log\frac{MH|S||A|}{\delta} + \sum_{m=1}^{M}\left(\sum_{h=1}^{h_m} c(s_h^m, a_h^m) + c_f(s_{h+1}^m)\mathbb{I}\{h_m = H\}\right)$$

$$\leq 10\alpha H^5 B_\star^{-2}|S|^2|A|\log\frac{MH|S||A|}{\delta} + 2\sum_{m=1}^{M}\mathbb{E}\left[\sum_{h=1}^{h_m} c(s_h^m, a_h^m) + c_f(s_{h+1}^m)\mathbb{I}\{h_m = H\}\,\Big|\,\bar{U}^m\right]$$

$$\leq O\left(H^5 B_\star^{-2}|S|^2|A|\log\frac{MH|S||A|}{\delta} + B_\star M\right),$$

where the second inequality follows since every state-action pair becomes known after the number of visits is $\alpha H^4 B_\star^{-2}|S|\log\frac{MH|S||A|}{\delta}$, and the last one by Lemma B.10. $\square$

**Lemma B.8** (Bound on Variance Term). *Conditioning on the good event, it holds that*

$$\sum_{m=1}^{M}\sum_{h=1}^{H} \frac{\sqrt{\mathrm{Var}_{P(\cdot|s_h^m,a_h^m)}(J_{h+1}^{\pi^m})}}{\sqrt{n^{m-1}(s_h^m, a_h^m)}} \leq O\left(\sqrt{B_\star^2|S||A|M\log(MH)} + H^3 B_\star^{-1}|S|^{3/2}|A|\log\frac{MH|S||A|}{\delta}\right).$$

*Proof.* Applying Cauchy-Schwartz inequality we get

$$\sum_{m=1}^{M}\sum_{h=1}^{H}\frac{\sqrt{\mathrm{Var}_{P(\cdot|s_h^m,a_h^m)}(J_{h+1}^{\pi^m})}}{\sqrt{n^{m-1}(s_h^m,a_h^m)\vee 1}} \le \sum_{m=1}^{M}\sum_{h=1}^{H}\frac{\sqrt{\mathrm{Var}_{P(\cdot|s_h^m,a_h^m)}(J_{h+1}^{\pi^m})}}{\sqrt{n^{m-1}(s_h^m,a_h^m)\vee 1}}\mathbb{I}\{n^{m-1}(s_h^m,a_h^m)\ge H\}+2H^2|S||A|$$

$$\le \sqrt{\sum_{m=1}^{M}\sum_{h=1}^{H}\mathrm{Var}_{P(\cdot|s_h^m,a_h^m)}(J_{h+1}^{\pi^m})}\sqrt{\sum_{m=1}^{M}\sum_{h=1}^{H}\frac{1}{n^{m-1}(s_h^m,a_h^m)\vee 1}\mathbb{I}\{n^{m-1}(s_h^m,a_h^m)\ge H\}}+2H^2|S||A|$$

$$\le \sqrt{\sum_{m=1}^{M}\sum_{h=1}^{H}\mathrm{Var}_{P(\cdot|s_h^m,a_h^m)}(J_{h+1}^{\pi^m})}\sqrt{3|S||A|\log(MH)}+2H^2|S||A| \qquad \text{(Lemma B.12)}$$

$$\le \sqrt{2\sum_{m=1}^{M}\mathbb{E}\left[\sum_{h=1}^{H}\mathrm{Var}_{P(\cdot|s_h^m,a_h^m)}(J_{h+1}^{\pi^m})\mid \bar{U}^m\right]+16H^3 L_M}\sqrt{3|S||A|\log(MH)}+2H^2|S||A|$$

$$\text{(Event } E^{\mathrm{Var}} \text{ holds)}$$

$$\le 3\sqrt{\sum_{m=1}^{M}\mathbb{E}\left[\sum_{h=1}^{H}\mathrm{Var}_{P(\cdot|s_h^m,a_h^m)}(J_{h+1}^{\pi^m})\mid \bar{U}^m\right]}\sqrt{|S||A|\log(MH)}$$

$$+7\sqrt{|S||A|H^3\log(MH)L_M}+2H^2|S||A| \qquad (\sqrt{a+b}\le\sqrt{a}+\sqrt{b})$$

$$\overset{(a)}{=} 3\sqrt{\sum_{m=1}^{M}\mathbb{E}\left[\left(\sum_{h=1}^{H}c(s_h^m,a_h^m)+c_f(s_{h+1}^m)-J_1^{\pi^m}(s_1)\right)^2\mid \bar{U}^m\right]}\sqrt{|S||A|\log(MH)}$$

$$+7\sqrt{|S||A|H^3\log(MH))L_m}+2H^2|S||A|$$

$$\overset{(b)}{\le} 3\sqrt{\sum_{m=1}^{M}\mathbb{E}\left[\left(\sum_{h=1}^{H}c(s_h^m,a_h^m)+c_f(s_{h+1}^m)\right)^2\mid \bar{U}^m\right]}\sqrt{|S||A|\log(MH)}+9H^2|S||A|L_M$$

$$\le O\left(\sqrt{B_\star^2|S||A|M\log(MH)}+H^3 B_\star^{-1}|S|^{3/2}|A|\log\frac{MH|S||A|}{\delta}\right),$$

where (a) is by law of total variance Azar et al. [2017], see Lemma B.14, (b) is because the variance is bounded by the second moment, and the last inequality is by Lemma B.9. □

## B.5   Bounds on the second moment

**Lemma B.9.** *Conditioning on the good event, it holds that*

$$\sum_{m=1}^{M}\mathbb{E}\left[\left(\sum_{h=1}^{H}c(s_h^m,a_h^m)+c_f(s_{h+1}^m)\right)^2\mid \bar{U}^m\right]\le O\left(B_\star^2 M+H^6 B_\star^{-2}|S|^2|A|\log\frac{MH|S||A|}{\delta}\right).$$

*Proof.* Denote by $h_m$ the last time step before reaching an unknown state-action pair (or $H$ if it was not reached). By the event $E^{Sec1}$ we have

$$\sum_{m=1}^{M}\mathbb{E}\left[\left(\sum_{h=1}^{H}c(s_h^m,a_h^m)+c_f(s_{h+1}^m)\right)^2 \;\middle|\; \bar{U}^m\right] \le 2\sum_{m=1}^{M}\left(\sum_{h=1}^{H}c(s_h^m,a_h^m)+c_f(s_{h+1}^m)\right)^2+62H^4L_M$$

$$\le 4\sum_{m=1}^{M}\left(\sum_{h=h_m+1}^{H}c(s_h^m,a_h^m)+c_f(s_{h+1}^m)\mathbb{I}\{h_m\ne H\}\right)^2+62H^4L_M$$

$$+4\sum_{m=1}^{M}\left(\sum_{h=1}^{h_m}c(s_h^m,a_h^m)+c_f(s_{h+1}^m)\mathbb{I}\{h_m=H\}\right)^2$$

$$\le 300\alpha H^6B_\star^{-2}|S|^2|A|\log\frac{MH|S||A|}{\delta}+4\sum_{m=1}^{M}\left(\sum_{h=1}^{h_m}c(s_h^m,a_h^m)+c_f(s_{h+1}^m)\mathbb{I}\{h_m=H\}\right)^2$$

$$\le 400\alpha H^6B_\star^{-2}|S|^2|A|\log\frac{MH|S||A|}{\delta}+4\sum_{m=1}^{M}\mathbb{E}\left[\left(\sum_{h=1}^{h_m}c(s_h^m,a_h^m)+c_f(s_{h+1}^m)\mathbb{I}\{h_m=H\}\right)^2 \;\middle|\; \bar{U}^m\right]$$

$$\le O\left(H^6B_\star^{-2}|S|^2|A|\log\frac{MH|S||A|}{\delta}+B_\star^2M\right),$$

where the third inequality follows since every state-action pair becomes known after the number of visits is $\alpha H^4B_\star^{-2}|S|\log\frac{MH|S||A|}{\delta}$, the forth inequality by event $E^{Sec2}$, and the last one by Lemma B.10. $\qquad\square$

**Lemma B.10.** *Let $m$ be an episode and $h_m$ be the last time step before an unknown state-action pair was reached (or $H$ if they were not reached). Further, denote by $C^m=\sum_{h=1}^{h_m}c(s_h^m,a_h^m)+c_f(s_{H+1}^m)\mathbb{I}\{h_m=H\}$ the cumulative cost in the episode until time $h_m$. Then, under the good event, $\mathbb{E}[C^m\mid\bar{U}^m]\le 3B_\star$ and $\mathbb{E}[(C^m)^2\mid\bar{U}^m]\le 2\cdot 10^4 B_\star^2$.*

*Proof.* Consider the following finite-horizon MDP $\mathcal{M}^m=(S\cup\{g\},A,P^m,H,c^m,c_f^m)$ that contracts unknown state-action pairs with a new goal state, i.e., $c^m(s,a)=c(s,a)\mathbb{I}\{s\ne g\}$ and $c_f^m(s)=c_f(s)\mathbb{I}\{s\ne g\}$ and

$$P_h^m(s'\mid s,a)=\begin{cases}0, & (s',\pi_{h+1}^m(s')) \text{ is unknown};\\ P(s'\mid s,a), & s'\ne g \text{ and } (s',\pi_{h+1}^m(s')) \text{ is known};\\ 1-\sum_{s''\in S}P_h^m(s''\mid s,a), & s'=g.\end{cases}$$

Denote by $J^m$ the cost-to-go function of $\pi^m$ in the MDP $\mathcal{M}^m$. Moreover, we slightly abuse notation to let $\widetilde{P}^m$ be the transition function induced by $\bar{P}^{m-1}$ in the MDP $\mathcal{M}^m$ similarly to $P^m$, and $\widetilde{J}^m$ the cost-to-go function of $\pi^m$ with respect to $\bar{P}^{m-1}$ (and cost function $\tilde{c}^m=\bar{c}^{m-1}-b_c^m-b_p^m$). By the value difference lemma (see, e.g., Shani et al., 2020), for every $s,h$ such that $(s,\pi_h^m(s))$ is known,

$$J_h^m(s)=\widetilde{J}_h^m(s)+\sum_{h'=h}^{H}\mathbb{E}\left[c^m(s_{h'},a_{h'})-\tilde{c}_{h'}^m(s_{h'},a_{h'})\mid s_h=s,P^m,\pi^m\right]$$

$$+\sum_{h'=h}^{H}\mathbb{E}\left[\left(P_{h'}^m(\cdot\mid s_{h'},a_{h'})-\widetilde{P}_{h'}^m(\cdot\mid s_{h'},a_{h'})\right)\cdot\widetilde{J}^m\mid s_h=s,P^m,\pi^m\right]$$

$$\le\widetilde{J}_h^m(s)+H\max_{\substack{(s,\pi_{h'}^m(s))\\ \text{known}}}|c(s,\pi_{h'}^m(s))-\tilde{c}_{h'}^m(s,\pi_{h'}^m(s))|+H\|\widetilde{J}^m\|_\infty\max_{\substack{(s,\pi_{h'}^m(s))\\ \text{known}}}\|P_{h'}^m(\cdot|s,\pi_{h'}^m(s))-\widetilde{P}_{h'}^m(\cdot|s,\pi_{h'}^m(s))\|_1$$

$$\le\widetilde{J}_h^m(s)+H\max_{\substack{(s,\pi_{h'}^m(s))\\ \text{known}}}|c(s,\pi_{h'}^m(s))-\tilde{c}_{h'}^m(s,\pi_{h'}^m(s))|$$

$$+2H\|\widetilde{J}^m\|_\infty\max_{\substack{(s,\pi_{h'}^m(s))\\ \text{known}}}\|P(\cdot|s,\pi_{h'}^m(s))-\bar{P}^{m-1}(\cdot|s,\pi_{h'}^m(s))\|_1$$

$$\le J_h^*(s)+H\max_{\substack{(s,\pi_{h'}^m(s))\\ \text{known}}}|c(s,\pi_{h'}^m(s))-\tilde{c}_{h'}^m(s,\pi_{h'}^m(s))|+2HB_\star\max_{\substack{(s,\pi_{h'}^m(s))\\ \text{known}}}\|P(\cdot|s,\pi_{h'}^m(s))-\bar{P}^{m-1}(\cdot|s,\pi_{h'}^m(s))\|_1,$$

where the last inequality follows by optimism and since $J_h^\star(s) \leq B_\star$. Thus, by Appendix B.2 (since all state-action pairs in $\mathcal{M}^m$ are known), we have that $J_h^m(s) \leq J_h^\star(s) + 2B_\star \leq 3B_\star$. Notice that $C^m$ is exactly the cost in the MDP $\mathcal{M}^m$, so $\mathbb{E}[C^m \mid \bar{U}^m] \leq 3B_\star$.

Similarly, we notice that $\mathbb{E}[(C^m)^2 \mid \bar{U}^m] = \mathbb{E}[(\widehat{C})^2]$, where $\widehat{C}$ is the cumulative cost in $\mathcal{M}^m$, and we override notation by denoting $\widehat{C} = \sum_{h=1}^{H} c(s_h, a_h) + c_f(s_{H+1})$. We split the time steps into $Q$ blocks as follows. We denote by $t_1$ the first time step in which we accumulated a total cost of at least $3B_\star$ (or $H + 1$ if it did not occur), by $t_2$ the first time step in which we accumulated a total cost of at least $3B_\star$ after $t_1$, and so on up until $t_Q = H + 1$. Then, the first block consists of time steps $t_0 = 1, \ldots, t_1 - 1$, the second block consists of time steps $t_1, \ldots, t_2 - 1$, and so on. Since $J_h^m(s) \leq 3B_\star$ we must have $c(s_h, a_h) \leq 3B_\star$ for all $h = 1, \ldots, H$ and thus in every such block the total cost is between $3B_\star$ and $6B_\star$. Thus,

$$\mathbb{E}\left[\left(\sum_{h=1}^{H} c(s_h, a_h) + c_f(s_{H+1})\right)^2\right] \geq \mathbb{E}\left[\sum_{h=1}^{H} c(s_h, a_h) + c_f(s_{H+1})\right]^2$$
$$= \mathbb{E}\left[\sum_{i=0}^{Q-1} \sum_{h=t_i}^{t_{i+1}-1} c(s_h, a_h) + c_f(s_{H+1})\right]^2$$
$$\geq \mathbb{E}[3B_\star Q]^2 = 9B_\star^2 \mathbb{E}[Q]^2,$$

by Jensen's inequality. On the other hand,

$$\mathbb{E}\left[\left(\sum_{h=1}^{H} c(s_h, a_h) + c_f(s_{H+1})\right)^2\right] = \mathbb{E}\left[\left(\sum_{h=1}^{H} c(s_h, a_h) + c_f(s_{H+1}) - J_1^m(s_1) + J_1^m(s_1)\right)^2\right]$$
$$\leq 2\mathbb{E}\left[\left(\sum_{h=1}^{H} c(s_h, a_h) + c_f(s_{H+1}) - J_1^m(s_1)\right)^2\right] + 2J_1^m(s_1)^2$$
$$\leq 2\mathbb{E}\left[\left(\sum_{i=0}^{Q-1} \sum_{h=t_i}^{t_{i+1}-1} c(s_h, a_h) - J_{t_i}^m(s_{t_i}) + J_{t_{i+1}}^m(s_{t_{i+1}})\right)^2\right] + 18B_\star^2$$
$$\overset{(a)}{=} 4\mathbb{E}\left[\sum_{i=0}^{Q-1} \left(\sum_{h=t_i}^{t_{i+1}-1} c(s_h, a_h) - J_{t_i}^m(s_{t_i}) + J_{t_{i+1}}^m(s_{t_{i+1}})\right)^2\right] + 18B_\star^2$$
$$\leq 4\mathbb{E}[Q \cdot (9B_\star)^2] + 18B_\star^2 \leq 324B_\star^2 \mathbb{E}[Q] + 18B_\star^2.$$

For (a) we used the fact that $\mathbb{E}[\sum_{h=t_i}^{t_{i+1}-1} c(s_h, a_h) - J_{t_i}(s_{t_i}) + J_{t_{i+1}}(s_{t_{i+1}})] = 0$ using the Bellman optimality equations and conditioned on all past randomness up until time $t_i$, and the fact that $t_{i+1}$ is a stopping time, in the following manner,

$$\mathbb{E}\left[\sum_{h=t_i}^{t_{i+1}-1} c(s_h, a_h) - J_{t_i}^m(s_{t_i}) + J_{t_{i+1}}^m(s_{t_{i+1}})\right] = \mathbb{E}\left[\sum_{h=t_i}^{t_{i+1}-1} c(s_h, a_h) - J_h^m(s_h) + J_{h+1}^m(s_{h+1})\right]$$
$$= \mathbb{E}\left[\sum_{h=t_i}^{t_{i+1}-1} \mathbb{E}\left[c(s_h, a_h) - J_h^m(s_h) + J_{h+1}^m(s_{h+1}) \mid s_h\right]\right]$$
$$= \mathbb{E}\left[\sum_{h=t_i}^{t_{i+1}-1} c(s_h, a_h) + \mathbb{E}\left[J_{h+1}^m(s_{h+1}) \mid s_h\right] - J_h^m(s_h)\right] = 0.$$

Thus, we have

$$9B_\star^2 \mathbb{E}[Q]^2 \leq 324B_\star^2 \mathbb{E}[Q] + 18B_\star^2,$$

and solving for $\mathbb{E}[Q]$ we obtain $\mathbb{E}[Q] \leq 37$, so

$$\mathbb{E}[(C^m)^2 \mid \bar{U}^m] = \mathbb{E}\left[\left(\sum_{h=1}^{H} \hat{c}(s_h, a_h) + \hat{c}_f(s_{H+1})\right)^2\right] \leq 2 \cdot 10^4 B_\star^2.$$

$\square$

**Lemma B.11** (Variance Difference is Upper Bounded by Value Difference). *Assume that the value at time step $h+1$ is optimistic, i.e., $\underline{J}_{h+1}^m(s) \le J_{h+1}^*(s)$ for all $s \in S$. Conditioning on the event $\cap_m E^{pv2}(m)$ it holds for all $(s,a) \in S \times A$ that*

$$\sqrt{2L_m} \frac{\left| \sqrt{\mathrm{Var}_{\bar{P}^{m-1}(\cdot|s,a)}(\underline{J}_{h+1}^m)} - \sqrt{\mathrm{Var}_{P(\cdot|s,a)}(J_{h+1}^*)} \right|}{\sqrt{n^{m-1}(s,a) \vee 1}} \le \frac{1}{\alpha} \mathbb{E}_{\bar{P}^{m-1}(\cdot|s,a)} \left[ J_{h+1}^*(s') - \underline{J}_{h+1}^m(s') \right] + \frac{(5 + \alpha/2)B_\star L_m}{n^{m-1}(s,a) \vee 1},$$

*for any $\alpha > 0$.*

*Proof.* Conditioning on $\cap_m E^{pv2}(m)$, the following relations hold.

$$\left| \sqrt{\mathrm{Var}_{\bar{P}^{m-1}(\cdot|s,a)}(\underline{J}_{h+1}^m)} - \sqrt{\mathrm{Var}_{P(\cdot|s,a)}(J_{h+1}^*)} \right| \le \left| \sqrt{\mathrm{Var}_{\bar{P}^{m-1}(\cdot|s,a)}(\underline{J}_{h+1}^m)} - \sqrt{\mathrm{Var}_{\bar{P}^{m-1}(\cdot|s,a)}(J_{h+1}^*)} \right|$$

$$+ \sqrt{\frac{12 B_\star^2 L_m}{n^{m-1}(s,a) \vee 1}}$$

$$\le \sqrt{\mathrm{Var}_{\bar{P}^{m-1}(\cdot|s,a)}(J_{h+1}^* - \underline{J}_{h+1}^m)} + \sqrt{\frac{12 B_\star^2 L_m}{n^{m-1}(s,a) \vee 1}}$$

$$\le \sqrt{\mathbb{E}_{\bar{P}^{m-1}} \left[ (J_{h+1}^*(s') - \underline{J}_{h+1}^m(s'))^2 \right]} + \sqrt{\frac{12 B_\star^2 L_m}{n^{m-1}(s,a) \vee 1}}$$

$$\le \sqrt{B_\star \mathbb{E}_{\bar{P}^{m-1}} \left[ J_{h+1}^*(s') - \underline{J}_{h+1}^m(s') \right]} + \sqrt{\frac{12 B_\star^2 L_m}{n^{m-1}(s,a) \vee 1}},$$

where the second inequality is by Lemma E.3, and the last relation holds since $J_{h+1}^*(s'), \underline{J}_{h+1}^m(s') \in [0, B_\star]$ (the first, by model assumption, and the second, by the update rule) and since $J_{h+1}^*(s') \ge \underline{J}_{h+1}^m(s')$ by the assumption the value is optimistic. Thus,

$$\sqrt{2L_m} \frac{\left| \sqrt{\mathrm{Var}_{\bar{P}^{m-1}(\cdot|s,a)}(\underline{J}_{h+1}^m)} - \sqrt{\mathrm{Var}_{P(\cdot|s,a)}(J_{h+1}^*)} \right|}{\sqrt{n^{m-1}(s,a)}} \le \sqrt{\mathbb{E}_{\bar{P}^{m-1}} \left[ J_{h+1}^*(s') - \underline{J}_{h+1}^m(s') \right]} \sqrt{\frac{2 B_\star L_m}{n^{m-1}(s,a) \vee 1}}$$

$$+ \frac{\sqrt{24} B_\star L_m}{n^{m-1}(s,a) \vee 1}$$

$$\le \frac{1}{\alpha} \mathbb{E}_{\bar{P}^{m-1}} \left[ J_{h+1}^*(s') - \underline{J}_{h+1}^m(s') \right] + \frac{(5 + \alpha/2) B_\star L_m}{n^{m-1}(s,a) \vee 1},$$

where the last inequality is by Young's inequality, $ab \le \frac{1}{\alpha} a^2 + \frac{\alpha}{4} b^2$. $\qquad\square$

## B.6 Useful results for reinforcement learning analysis

**Lemma B.12** (Cumulative Visitation Bound for Stationary MDP, e.g., Efroni et al., 2020, Lemma 23). *It holds that*

$$\sum_{m=1}^M \sum_{s,a} \mathbb{I}\{n^{m-1}(s,a) \ge H\} \frac{\sum_{h=1}^H \mathbb{I}\{s_h^m = s, a_h^m = a\}}{n^{m-1}(s,a) \vee 1} \le 3|S||A| \log(MH).$$

*Proof.* Recall that we recompute the optimistic policy only in the end of episodes in which the number of visits to some state-action pair was doubled. In this proof we refer to a sequence of consecutive episodes in which we did not perform a recomputation of the optimistic policy by the name of *epoch*. Let $E$ be the number of epochs and note that $E \le |S||A| \log(MH)$ because the number of visits to each state-action pair $(s,a)$ can be doubled at most $\log(MH)$ times. Next, denote by $\tilde{n}^e(s,a)$ the number of visits to $(s,a)$ until the end of epoch $e$ and by $\tilde{N}^e(s,a)$ the number of visits to $(s,a)$

during epoch $e$. The following relations hold for any fixed $(s, a)$ pair.

$$\sum_{m=1}^{M} \mathbb{I}\{n^{m-1}(s, a) \geq H\} \frac{\sum_{h=1}^{H} \mathbb{I}\{s_h^m = s, a_h^m = a\}}{n^{m-1}(s, a) \vee 1} =$$

$$= \sum_{e=1}^{E} \mathbb{I}\{\tilde{n}^{e-1}(s, a) \geq H\} \frac{\widetilde{N}^e(s, a)}{\tilde{n}^{e-1}(s, a)}$$

$$= \sum_{e=1}^{E} \mathbb{I}\{\tilde{n}^{e-1}(s, a) \geq H\} \frac{\widetilde{N}^e(s, a)}{\tilde{n}^e(s, a)} \frac{\tilde{n}^e(s, a)}{\tilde{n}^{e-1}(s, a)}$$

$$\leq 3 \sum_{e=1}^{E} \mathbb{I}\{\tilde{n}^{e-1}(s, a) \geq H\} \frac{\widetilde{N}^e(s, a)}{\tilde{n}^e(s, a)}$$

$$= 3 \sum_{e=1}^{E} \mathbb{I}\{\tilde{n}^{e-1}(s, a) \geq H\} \frac{\tilde{n}^e(s, a) - \tilde{n}^{e-1}(s, a)}{n^e(s, a)}$$

$$\leq 3 \sum_{e=1}^{E} \mathbb{I}\{\tilde{n}^{e-1}(s, a) \geq H\} \log \left( \frac{\tilde{n}^e(s, a)}{\tilde{n}^{e-1}(s, a)} \right)$$

$$\leq 3 \mathbb{I}\{\tilde{n}^E(s, a) \geq H\}(\log \tilde{n}^E(s, a) - \log(H))$$

$$\leq 3 \log\left(\tilde{n}^E(s, a) \vee 1\right),$$

where the first inequality follows since $\frac{\tilde{n}^e(s,a)}{\tilde{n}^{e-1}(s,a)} \leq \frac{2\tilde{n}^{e-1}(s,a)+H}{\tilde{n}^{e-1}(s,a)} \leq 3$ for $\tilde{n}^{e-1}(s, a) \geq H$, and the second inequality follows by the inequality $\frac{a-b}{a} \leq \log \frac{a}{b}$ for $a \geq b > 0$. Applying Jensen's inequality we conclude the proof:

$$\sum_{m=1}^{M} \sum_{s,a} \mathbb{I}\{n^{m-1}(s, a) \geq H\} \frac{\sum_{h=1}^{H} \mathbb{I}\{s_h^m = s, a_h^m = a\}}{n^{m-1}(s, a) \vee 1} \leq 3 \sum_{s,a} \log\left(\tilde{n}^E(s, a) \vee 1\right)$$

$$\leq 3|S||A| \log \left( \sum_{s,a} \tilde{n}^E(s, a) \right)$$

$$\leq 3|S||A| \log(MH).$$

$\square$

**Lemma B.13** (Transition Difference to Next State Expectation, Efroni et al., 2021, Lemma 28). *Let $Y \in \mathbb{R}^{|S|}$ be a vector such that $0 \leq Y(s) \leq 2H$ for all $s \in S$. Let $P_1$ and $P_2$ be two transition models and $n \in \mathbb{R}_+^{|S||A|}$. Let $\Delta P(\cdot \mid s, a) \in \mathbb{R}^{|S|}$ and $\Delta P(s'|s, a) = P_1(s'|s, a) - P_2(s'|s, a)$. Assume that*

$$\forall (s, a, s') \in S \times A \times S, h \in [H] : |\Delta P(s'|s, a)| \leq \sqrt{\frac{C_1 L_m P_1(s'|s, a)}{n(s, a) \vee 1}} + \frac{C_2 L_m}{n(s, a) \vee 1},$$

*for some $C_1, C_2 > 0$. Then, for any $\alpha > 0$.*

$$|\Delta P(\cdot \mid s, a) \cdot Y| \leq \frac{1}{\alpha} \mathbb{E}_{P_1(\cdot|s,a)}\left[Y(s')\right] + \frac{HL_m(2C_2 + \alpha|S|C_1/2)}{n(s, a) \vee 1}.$$

**Lemma B.14** (Law of Total Variance, e.g., Azar et al., 2017). *For any $\pi$ the following holds.*

$$\mathbb{E}\left[\sum_{h=1}^{H} \text{Var}_{P(\cdot|s_h,a_h)}(J_{h+1}^\pi) \mid \pi\right] = \mathbb{E}\left[\left(\sum_{h=1}^{H} c(s_h, a_h) + c_f(s_{H+1}) - J_1^\pi(s_1)\right)^2 \mid \pi\right].$$

# C   Extending the reduction to unknown $B_\star$

In this section we assume $B_\star \geq 1$ to simplify presentation, but the results work similarly for $B_\star < 1$. To handle unknown $B_\star$, we leverage techniques from the adversarial SSP literature [Rosenberg and Mansour, 2020, Chen and Luo, 2021] for learning the diameter of an SSP problem. Recall that the SSP-diameter $D$ [Tarbouriech et al., 2020] is defined as $D = \max_{s \in S} \min_{\pi:s \to A} T^\pi(s)$. So to compute $D$ we can find the optimal policy with respect to the constant cost function $c_1(s, a) = 1$, and compute its cost-to-go function. Rosenberg and Mansour [2020] utilize this observation to estimate the SSP-diameter. They show that one can estimate the expected time from a state $s$ to the goal state $g$ by running the `Bernstein-SSP` algorithm of Rosenberg et al. [2020] with unit costs for $L = \widetilde{O}(D^2|S|^2|A|)$ episodes and setting the estimator to be the average cost per episode times 10.

Inspired by their approach, we use the `Bernstein-SSP` algorithm on the the actual costs, in order to estimate the expected cost of the optimal policy. Although `Bernstein-SSP` suffers from sub-optimal regret, we run it only for a small number of episodes and therefore we will only suffer from a slightly larger additive factors in our regret bound, but keep minimax optimal regret for large enough $K$.

By similar proofs to Lemmas 26 and 27 from Rosenberg and Mansour [2020, Appendix J], we can show that the cost-to-go from state $s$ can be estimated up to a constant multiplicative factor by running `Bernstein-SSP` for $L = \widetilde{O}(T_\star^2|S|^2|A|)$ episodes. This is demonstrated in the following lemma, where the upper bound follows from the regret guarantees of `Bernstein-SSP` and the lower bound follows from concentration arguments (and noticing that the regret is minimized by playing the optimal policy, but even then it is not zero).

**Lemma C.1.** *Let $s \in S$ and $L \geq 2400 T_\star^2|S|^2|A| \log^3 \frac{KT_\star|S||A|}{\delta}$. Run* `Bernstein-SSP` *with initial state $s$ for $L$ episodes and denote by $\widetilde{B}_s$ the average cost per episode times 10. Then, with probability $1 - \delta$,*

$$J^{\pi^\star}(s) \leq \widetilde{B}_s \leq O(B_\star).$$

Thus, we use the first $L$ visits to each state in order to estimate its cost-to-go. A state which was visited at least $L$ times will be called $B_\star$-*known*, and otherwise $B_\star$-*unknown* (not to be confused with our previous definition of known state-action pair). To that end, we split the total time steps into $E$ epochs. In epoch $e$, we apply our reduction to a virtual MDP $\mathcal{M}^e$ that is identical to $\mathcal{M}$ in $B_\star$-known states, but turns $B_\star$-unknown states into zero-cost sinks (like the goal state). For every state $s \in S$ we maintain a `Bernstein-SSP` algorithm $\mathcal{B}_s$. Every time we reach a $B_\star$-unknown state $s$, we run an episode of $\mathcal{B}_s$ until the goal is reached.

Note that in the virtual MDP $\mathcal{M}^e$ we can compute an upper bound on the optimal cost-to-go using our estimates. Epoch $e$ ends once some $B_\star$-unknown state $s$ is visited $L$ times and thus becomes $B_\star$-known. Therefore the number of epochs $E$ is bounded by $|S|$. The important change, introduced by Chen and Luo [2021], is to not completely initialize our finite-horizon algorithm $\mathcal{A}$ in the beginning of a new epoch as this leads to an extra $|S|$ factor in the regret. Instead, algorithm $\mathcal{A}$ inherits the experience (i.e., visit counters and accumulated costs) of the previous epoch in $B_\star$-known states.

The reduction without knowledge of $B_\star$ is presented in Algorithm 4, and next we prove that it maintains the same regret bound up to a slightly larger additive factor.

**Theorem C.2.** *Let $\mathcal{A}$ be an admissible algorithm for regret minimization in finite-horizon MDPs and denote its regret in $M$ episodes by $\widehat{\mathcal{R}}_\mathcal{A}(M)$. Then, running Algorithm 4 with $\mathcal{A}$ ensures that, with probability at least $1 - 2\delta$,*

$$R_K \leq \widehat{\mathcal{R}}_\mathcal{A}\left( 4K + 4 \cdot 10^4|S||A|\omega_\mathcal{A} \log \frac{KT_\star|S||A|\omega_\mathcal{A}}{\delta} + 4 \cdot 10^4 T_\star^2|S|^3|A| \log^3 \frac{KT_\star|S||A|}{\delta} \right)$$

$$+ O\left( B_\star\sqrt{K \log \frac{KT_\star|S||A|\omega_\mathcal{A}}{\delta}} + T_\star\omega_\mathcal{A}|S||A| \log^2 \frac{KT_\star|S||A|\omega_\mathcal{A}}{\delta} + T_\star^3|S|^3|A| \log^4 \frac{KT_\star|S||A|}{\delta} \right),$$

*where $\omega_\mathcal{A}$ is a quantity that depends on the algorithm $\mathcal{A}$ and on $|S|, |A|, H$.*

Using the reduction with the `ULCVI` algorithm, we can again obtain optimal regret for SSP.

**Theorem C.3.** *Running the reduction in Algorithm 4 with the finite-horizon regret minimization algorithm* ULCVI *ensures, with probability at least* $1 - 2\delta$,

$$R_K = O\left( B_\star \sqrt{|S||A|K} \log \frac{KT_\star|S||A|}{\delta} + T_\star^5|S|^2|A| \log^6 \frac{KT_\star|S||A|}{\delta} + T_\star^3|S|^3|A| \log^4 \frac{KT_\star|S||A|}{\delta} \right).$$

---

**Algorithm 4** REDUCTION FROM SSP TO FINITE-HORIZON MDP WITH UNKNOWN $B_\star$

---
1: **input:** state space $S$, action space $A$, initial state $s_{\text{init}}$, goal state $g$, confidence parameter $\delta$, number of episodes $K$, bound on the expected time of the optimal policy $T_\star$ and algorithm $\mathcal{A}$ for regret minimization in finite-horizon MDPs.
2: **initialize** a Bernstein-SSP algorithm $\mathcal{B}_s$ with initial state $s$ and confidence parameter $\delta/|S|$ for every $s \in S$.
3: set $L = 10^4 T_\star^2 |S|^2 |A| \log^3 \frac{KT_\star|S||A|}{\delta}$, $S_{\text{known}}^1 = \{s_{\text{init}}\}$ and $N_f(s) = L\mathbb{I}\{s = s_{\text{init}}\}$ for every $s \in S$.
4: run $\mathcal{B}_{s_{\text{init}}}$ for $L$ episodes and set $\widetilde{B}_{s_{\text{init}}}$ to be the average cost per episode times 10.
5: **initialize** $\mathcal{A}$ with state space $\widehat{S} = S \cup \{g\}$, action space $A$, horizon $H = 8T_\star \log(8K)$, confidence parameter $\frac{\delta}{4|S|}$, terminal costs $\hat{c}_f(s) = 8\mathbb{I}\{s = s_{\text{init}}\}\widetilde{B}_{s_{\text{init}}}$ and bound on the expected cost of the optimal policy $9\widetilde{B}_{s_{\text{init}}}$.
6: **initialize** intervals counter $m \leftarrow 0$, time steps counter $t \leftarrow 1$ and epochs counter $e \leftarrow 1$.
7: **for** $k = L + 1, \dots, K$ **do**
8:     set $s_t \leftarrow s_{\text{init}}$.
9:     **while** $s_t \neq g$ **do**
10:         set $m \leftarrow m + 1$, feed initial state $s_t$ to $\mathcal{A}$ and obtain policy $\pi^m = \{\pi_h^m : \widehat{S} \to A\}_{h=1}^H$.
11:         **for** $h = 1, \dots, H$ **do**
12:             play action $a_t = \pi_h^m(s_t)$, suffer cost $C_t \sim c(s_t, a_t)$, and set $s_h^m = s_t, a_h^m = a_t, C_h^m = C_t$.
13:             observe next state $s_{t+1} \sim P(\cdot \mid s_t, a_t)$ and set $t \leftarrow t + 1$.
14:             **if** $s_t = g$ or $s_t \notin S_{\text{known}}^e$ **then**
15:                 pad trajectory to be of length $H$ and BREAK.
16:             **end if**
17:         **end for**
18:         set $s_{H+1}^m = s_t$.
19:         feed trajectory $U^m = (s_1^m, a_1^m, \dots, s_H^m, a_H^m, s_{H+1}^m)$ and costs $\{C_h^m\}_{h=1}^H$ to $\mathcal{A}$.
20:         **if** $s_t \notin S_{\text{known}}^e$ **then**
21:             set $N_f(s_t) \leftarrow N_f(s_t) + 1$ and run an episode of $\mathcal{B}_{s_t}$.
22:             **if** $N_f(s_t) = L$ **then**
23:                 set $e \leftarrow e + 1$ and $S_{\text{known}}^e \leftarrow S_{\text{known}}^{e-1} \cup \{s_t\}$.
24:                 set $\widetilde{B}_{s_t}$ to be the average cost per episode of $\mathcal{B}_{s_t}$ times 10.
25:                 **reinitialize** $\mathcal{A}$ by updating the terminal costs as $\hat{c}_f(s) = 8\mathbb{I}\{s \in S_{\text{known}}^e\} \max_{\tilde{s} \in S_{\text{known}}^e} \widetilde{B}_{\tilde{s}}$, updating the bound on the expected cost of the optimal policy $9 \max_{\tilde{s} \in S_{\text{known}}^e} \widetilde{B}_{\tilde{s}}$ and deleting the history of $\mathcal{A}$ only in state $s_t$.
26:             **end if**
27:         **end if**
28:     **end while**
29: **end for**

---

## C.1 Proof of Theorem C.2

We follow the analysis of the known $B_\star$ case under the event that Lemma C.1 holds for all states (which happens with probability at least $1 - \delta$), i.e., $J^{\pi^\star}(s) \leq \widetilde{B}_s \leq O(B_\star)$ for every $s \in S$. We start by decomposing the regret similarly to Lemma 4.1. Note that now there is an additional term that comes from the regret of the $|S|$ Bernstein-SSP algorithms that are used to estimate $B_\star$.

**Lemma C.4.** *For $H = 8T_\star \log(8K)$, we have the following bound on the regret of Algorithm 4:*

$$R_K \leq \widehat{\mathcal{R}}_{\mathcal{A}}(M) + \sum_{m=1}^M \left( \sum_{h=1}^H C_h^m + \hat{c}_f(s_{H+1}^m) - \widehat{J}_1^{\pi^m}(s_1^m) \right) + O\left( T_\star^2 B_\star |S|^3 |A| \log^3 \frac{KT_\star|S||A|}{\delta} \right), \quad (14)$$

*where M is the total number of intervals.*

**Remark 6.** Note that now each interval is considered in the context of the current epoch, i.e., the current $B_\star$-known states. The finite-horizon cost-to-go $\widehat{J}^{\pi^m}$ is with respect to the MDP of $B_\star$-known states. Moreover, for interval $m$ that ends in a $B_\star$-unknown state, the last state in the trajectory $s_{H+1}^m$ will be a $B_\star$-unknown state and the length of the interval may be shorter than $H$ (just like intervals that end in the goal state).

*Proof.* Every interval ends either in the goal state, in a $B_\star$-known state or in a $B_\star$-unknown state. The first two cases are similar to the proof of Lemma 4.1 because our estimates $\widetilde{B}_s$ in all $B_\star$-known states $s$ are upper bounds on $J^{\pi^\star}(s)$. Importantly, we do not initialize $\mathcal{A}$ in the end of an epoch and this allows us to get its regret bound without an extra $|S|$ factor. The reason is that $\mathcal{A}$ is an admissible (and thus optimistic) algorithm, so it operates based on the observations it collected. Another important note is that the cost in the virtual MDP $\mathcal{M}^e$ is always bounded by the cost in the actual MDP $\mathcal{M}$.

We now focus on the last case. Recall that if interval $m$ ends in a $B_\star$-unknown state $s$, then the terminal cost is 0 and we run an episode of the Bernstein-SSP algorithm $\mathcal{B}_s$. Thus, the excess cost of running Bernstein-SSP algorithms is bounded by $|S|$ times the Bernstein-SSP regret plus $|S|B_\star L$, i.e., we can bound it as follows

$$|S|B_\star L + O\left(B_\star^{3/2}|S|^2\sqrt{|A|L}\log\frac{KT_\star|S||A|}{\delta} + T_\star^{3/2}|S|^3|A|\log^2\frac{KT_\star|S||A|}{\delta}\right).$$

To finish the proof we plug in the definition of $L$. □

Next, we bound the number of intervals. Again, we get a similar bound to Lemma 4.3 but with an additional term for all the intervals that ended in a $B_\star$-unknown state (there are at most $|S|L$ such intervals).

**Lemma C.5.** *Assume that the reduction is performed using an admissible algorithm $\mathcal{A}$. Then, with probability at least $1 - 3\delta/8$,*

$$M \leq 4\left(K + 10^4|S||A|\omega_\mathcal{A}\log\frac{KT_\star|S||A|\omega_\mathcal{A}}{\delta} + 10^4 T_\star^2|S|^3|A|\log^3\frac{KT_\star|S||A|}{\delta}\right).$$

*Proof.* The proof is based on the claim that in every interval there is a probability of at least $1/2$ that the agent reaches either the goal state, an unknown state-action pair or a $B_\star$-unknown state. This is proved similarly to Lemma A.3 since we can look at the MDP of $B_\star$-known states, and then the claim of Lemma A.3 is equivalent to reaching either the goal state, an unknown state-action pair or a $B_\star$-unknown state.

With this claim the proof follows easily by following the proof of Lemma 4.3. We simply define $X^m$ to be 1 if an unknown state-action pair or the goal or a $B_\star$-unknown state were reached during interval $m$ (and 0 otherwise). Then, we have

$$\sum_{m=1}^{M} X^m \leq K + |S||A|\omega_\mathcal{A}\log\frac{MH|S||A|}{\delta} + |S|L,$$

which implies the Lemma following the same argument based on Freedman's inequality. □

Finally, we bound the deviation of the actual cost in each interval from its expected value. The proof is exactly the same as Lemma 4.2. The second moment of the accumulated cost until reaching the goal, an unknown state-action pair or a $B_\star$-unknown state is of order $B_\star^2$, and therefore in almost all intervals (except for a finite number) the accumulated cost will be of order $B_\star$ with high probability (in other intervals the cost is trivially bounded by $H + O(B_\star)$).

**Lemma C.6.** *Assume that the reduction is performed using an admissible algorithm $\mathcal{A}$. Then, the following holds with probability at least $1 - 3\delta/8$,*

$$\sum_{m=1}^{M}\left(\sum_{h=1}^{H} C_h^m + \hat{c}_f(s_{H+1}^m) - \widehat{J}_1^{\pi^m}(s_1^m)\right) = O\left(B_\star\sqrt{M\log\frac{M}{\delta}} + (H + B_\star)\omega_\mathcal{A}|S||A|\log\frac{MKT_\star|S||A|}{\delta}\right)$$

$$+ O\left((H + B_\star)T_\star^2|S|^3|A|\log^3\frac{KT_\star|S||A|}{\delta}\right).$$

The proof of the theorem is finished by combining Lemmas C.4 to C.6 together with the guarantees of the admissible algorithm $\mathcal{A}$ and Lemma C.1, similarly to Theorem 3.1.

# D Lower bound

In this section we prove Theorem 2.3 which lower bounds the expected regret of any learning algorithm for the case $B_\star < 1$. It complements the lower bound found in Rosenberg et al. [2020] for the case $B_\star \geq 1$.

By Yao's minimax principle, in order to derive a lower bound on the learner's regret, it suffices to show a distribution over MDP instances that forces any deterministic learner to suffer a regret of $\Omega(\sqrt{B_\star|S||A|K})$ in expectation.

To construct this distribution, we follow Rosenberg et al. [2020] with a few modifications. We initially consider the simpler setting with two states: an initial state and the goal state. We now embed a hard MAB instance into our problem where the optimal action has an expected cost of $B_\star$. To that end, consider a distribution over MDPs where a special action $a^\star$ is chosen a-priori uniformly at random. Then, all actions lead to the goal state $g$ with probability 1. The cost $C_k(s_{\text{init}}, a^\star)$ chosen at episode $k$ is 1 w.p. $B_\star$ and 0 otherwise. The cost of any other action $a \neq a^\star$ is 1 w.p. $B_\star + \epsilon$ and 0 otherwise, where $\epsilon \in (0, 1/8)$ is a constant to be determined. Thus the optimal policy will always play $a^\star$ and we have $J^{\pi^\star}(s_{\text{init}}) = B_\star$.

Fix any deterministic learning algorithm, we shall now quantify the regret of the learner in terms of the number of times that it plays $a^\star$. Indeed, we have that the optimal cost is $B_\star$, and the learner loses $\epsilon$ in the regret each time she plays an action other than $a^\star$. Therefore,

$$\mathbb{E}[R_K] \geq \epsilon \cdot (K - \mathbb{E}[N]),$$

where $N$ is the number of times $a^\star$ was chosen in $s_{\text{init}}$.

We now introduce an additional distribution of the costs which denote by $\mathbb{P}_{\text{unif}}$. $\mathbb{P}_{\text{unif}}$ is identical to the distribution over the costs defined above, and denoted by $\mathbb{P}$, except that $\mathbb{P}[C_k(s_{\text{init}}, a) = 1] = B_\star + \epsilon$ for all actions $a \in A$ regardless of the choice of $a^\star$. We denote expectations over $\mathbb{P}_{\text{unif}}$ by $\mathbb{E}_{\text{unif}}$, and expectations over $\mathbb{P}$ by $\mathbb{E}$. The following lemma uses standard lower bound techniques used for multi-armed bandits (see, e.g., Jaksch et al., 2010, Theorem 13) to bound the difference in the expectation of $N$ when the learner plays in $\mathbb{P}$ compared to when it plays in $\mathbb{P}_{\text{unif}}$.

**Lemma D.1.** *Suppose that $B_\star \leq \frac{1}{2}$. Denote by $\mathbb{P}_{\text{unif},a}$, $\mathbb{E}_{\text{unif},a}$, $\mathbb{P}_a$, $\mathbb{E}_a$ the distributions and expectations defined above conditioned on $a^\star = a$. For any deterministic learner we have that $\mathbb{E}_a[N] \leq \mathbb{E}_{\text{unif},a}[N] + \epsilon K \sqrt{\mathbb{E}_{\text{unif},a}[N]/B_\star}$.*

*Proof.* Fix any deterministic learner. Let us denote by $C^{(k)}$ the sequence of costs observed by the learner up to episode $k$ and including. Now, as $N \leq K$ and the fact that $N$ is a deterministic function of $C^{(K)}$, $\mathbb{E}_a[N] \leq \mathbb{E}_{\text{unif},a}[N] + K \cdot \text{TV}(\mathbb{P}_{\text{unif},a}[C^{(K)}], \mathbb{P}[C^{(K)}])$, and Pinsker's inequality yields

$$\text{TV}(\mathbb{P}_{\text{unif},a}[C^{(K)}], \mathbb{P}[C^{(K)}]) \leq \sqrt{\frac{1}{2}\text{KL}(\mathbb{P}_{\text{unif},a}[C^{(K)}] \parallel \mathbb{P}_a[C^{(K)}])}. \tag{15}$$

Next, the chain rule of the KL divergence obtains

$$\text{KL}(\mathbb{P}_{\text{unif},a}[C^{(K)}] \parallel \mathbb{P}_a[C^{(K)}])$$
$$= \sum_{k=1}^{K} \sum_{C^{(k)}} \mathbb{P}_{\text{unif},a}[C^{(k)}] \cdot \text{KL}(\mathbb{P}_{\text{unif},a}[C_k(s_{\text{init}}, a_k) \mid C^{(k)}] \parallel \mathbb{P}_a[C_k(s_{\text{init}}, a_k) \mid C^{(k)}]),$$

where $a_k$ is the action chosen by the learner at episode $k$. (Recall that after which the model transition to the goal state and the episode ends.)

Observe that at any episode, since the learning algorithm is deterministic, the learner chooses an action given $C^{(k)}$ regardless of whether $C^{(k)}$ was generated under $\mathbb{P}$ or under $\mathbb{P}_{\text{unif},a}$. Thus, the $\text{KL}(\mathbb{P}_{\text{unif},a}[C_k(s_{\text{init}}, a_k) \mid C^{(k)}] \parallel \mathbb{P}_a[C_k(s_{\text{init}}, a_k) \mid C^{(k)}])$ is zero if $a_k \neq a_\star$, and otherwise

$$\text{KL}(\mathbb{P}_{\text{unif},a}[C_k(s_{\text{init}}, a_k) \mid C^{(k)}] \parallel \mathbb{P}_a[C_k(s_{\text{init}}, a_k) \mid C^{(k)}])$$
$$= (B_\star + \epsilon) \log\left(1 + \frac{\epsilon}{B_\star}\right) + (1 - B_\star - \epsilon) \log\left(1 - \frac{\epsilon}{1 - B_\star}\right)$$
$$\leq \frac{\epsilon^2}{B_\star(1 - B_\star)},$$

where we used that $\log(1 + x) \le x$ for all $x > -1$, and since we assume $B_\star \le \frac{1}{2}$ and $\epsilon < \frac{1}{8}$ that imply $-\epsilon/(1 - B_\star) \ge -\frac{1}{4} > -1$. Plugging the above back into Eq. (15) and using $B_\star \le \frac{1}{2}$ gives the lemma. $\quad\square$

In the following result, we combine the lemma above with standard techniques from lower bounds of multi-armed bandits (see Auer et al., 2002 for example).

**Theorem D.2.** *Suppose that* $B_\star \le \frac{1}{2}$, $\epsilon \in (0, \frac{1}{8})$ *and* $|A| \ge 2$. *For the problem described above we have that*

$$\mathbb{E}[R_K] \ge \epsilon K \left( \frac{1}{2} - \epsilon \sqrt{\frac{K}{|A|B_\star}} \right).$$

*Proof of Theorem D.2.* Note that as under $\mathbb{P}_{\text{unif}}$ the cost distributions of all actions are identical. Denote by $N_a$ the number of times that the learner chooses action $a$ in $s_{\text{init}}$. Therefore,

$$\sum_{a \in A} \mathbb{E}_{\text{unif},a}[N] = \sum_{a \in A} \mathbb{E}_{\text{unif}}[N_a] = \mathbb{E}_{\text{unif}} \left[ \sum_{a \in A} N_a \right] = K. \tag{16}$$

Recall that $a^\star$ is sampled uniformly at random before the game starts. Then,

$$\begin{aligned}
\mathbb{E}[R_K] &= \frac{1}{|A|} \sum_{a \in A} \mathbb{E}_a[R_K] \\
&\ge K - \frac{1}{|A|} \sum_{a \in A} \mathbb{E}_a[N] \\
&\ge K - \frac{1}{|A|} \sum_{a \in A} \left( \mathbb{E}_{\text{unif},a}[N] + \epsilon K \sqrt{\mathbb{E}_{\text{unif},a}[N]/B_\star} \right) && \text{(Lemma D.1)} \\
&\ge K - \frac{1}{|A|} \sum_{a \in A} \mathbb{E}_{\text{unif},a}[N] + \epsilon K \sqrt{\frac{1}{|A|B_\star} \sum_{a \in A} \mathbb{E}_{\text{unif},a}[N]} && \text{(Jensen's inequality)} \\
&= K - \frac{K}{|A|} + \epsilon K \sqrt{\frac{K}{|A|B_\star}}, && \text{(Eq. (16))}
\end{aligned}$$

The theorem follows from $|A| \ge 2$ and by rearranging. $\quad\square$

*Proof of Theorem 2.3.* Consider the following MDP. Let $S$ be the set of states disregarding $g$. The initial state is sampled uniformly at random from $S$. Each $s \in S$ has its own special action $a_s^\star$. All actions transition to the goal state with probability 1. The cost $C_k(s, a)$ of action $a \ne a_s^\star$ in episode $k$ and state $s$ is 1 with probability $B_\star + \epsilon$ and 0 otherwise. The cost of $C_k(s, a_s^\star)$ is 1 with probability $B_\star$ and 0 otherwise.

Note that for each $s \in S$, the learner is faced with a simple problem as the one described above from which it cannot learn about from other states $s' \ne s$. Therefore, we can apply Theorem D.2 for each $s \in S$ separately and lower bound the learner's expected regret the sum of the regrets suffered at each $s \in S$, which would depend on the number of times $s \in S$ is drawn as the initial state. Since the states are chosen uniformly at random there are many states (constant fraction) that are chosen $\Theta(K/|S|)$ times. Summing the regret bounds of Theorem D.2 over only these states and choosing $\epsilon$ appropriately gives the sought-after bound.

Denote by $K_s$ the number of episodes that start in each state $s \in S$.

$$\mathbb{E}[R_K] \ge \sum_{s \in S} \mathbb{E} \left[ \epsilon K_s \left( \frac{1}{2} - \epsilon \sqrt{\frac{K_s}{|A|B_\star}} \right) \right] = \frac{\epsilon K}{2} - \epsilon^2 \sqrt{\frac{1}{|A|B_\star}} \sum_{s \in S} \mathbb{E}[K_s^{3/2}]. \tag{17}$$

Applying Cauchy-Schwartz inequality gives

$$\begin{aligned}
\sum_{s \in S} \mathbb{E}[K_s^{3/2}] &\le \sum_{s \in S} \sqrt{\mathbb{E}[K_s]} \sqrt{\mathbb{E}[K_s^2]} = \sum_{s \in S} \sqrt{\mathbb{E}[K_s]} \sqrt{\mathbb{E}[K_s]^2 + \text{Var}[K_s]} \\
&= \sum_{s \in S} \sqrt{\frac{K}{|S|}} \sqrt{\frac{K^2}{|S|^2} + \frac{K}{|S|} \left( 1 - \frac{1}{|S|} \right)} \le K \sqrt{\frac{2K}{|S|}},
\end{aligned}$$

where we have used the expectation and variance formulas of the Binomial distribution. The lower bound is now given by applying the inequality above in Eq. (17) and choosing $\epsilon = \frac{1}{8}\sqrt{B_\star|A||S|/K}$. $\qquad\square$

# E  General useful results

**Lemma E.1** (Freedman's Inequality). *Let $\{X_t\}_{t\geq 1}$ be a real valued martingale difference sequence adapted to a filtration $\{F_t\}_{t\geq 0}$. If $|X_t| \leq R$ a.s. then for any $\eta \in (0, 1/R), T \in \mathbb{N}$ it holds with probability at least $1 - \delta$,*

$$\sum_{t=1}^{T} X_t \leq \eta \sum_{t=1}^{T} \mathbb{E}[X_t^2 | F_{t-1}] + \frac{\log(1/\delta)}{\eta}.$$

**Lemma E.2** (Consequences of Freedman's Inequality for Bounded and Positive Sequence of Random Variables, e.g., Efroni et al., 2021, Lemma 27). *Let $\{Y_t\}_{t\geq 1}$ be a real valued sequence of random variables adapted to a filtration $\{F_t\}_{t\geq 0}$. Assume that for all $t \geq 1$ it holds that $0 \leq Y_t \leq C$ a.s., and $T \in \mathbb{N}$. Then, each of the following inequalities hold with probability at least $1 - \delta$.*

$$\sum_{t=1}^{T} \mathbb{E}[Y_t | F_{t-1}] \leq \left(1 + \frac{1}{2C}\right) \sum_{t=1}^{T} Y_t + 2(2C+1)^2 \log \frac{1}{\delta}$$

$$\sum_{t=1}^{T} Y_t \leq 2 \sum_{t=1}^{T} \mathbb{E}[Y_t | F_{t-1}] + 4C \log \frac{1}{\delta}.$$

**Lemma E.3** (Standard Deviation Difference, e.g., Zanette and Brunskill, 2019). *Let $V_1, V_2 : S \to \mathbb{R}$ be fixed mappings. Let $P(s)$ be a probability measure over the state space. Then, $\sqrt{\mathrm{Var}(V_1)} - \sqrt{\mathrm{Var}(V_2)} \leq \sqrt{\mathrm{Var}(V_1 - V_2)}$.*