# OpenReview forum: "Minimax Regret for Stochastic Shortest Path"
_NeurIPS.cc/2021/Conference — NeurIPS 2021 Poster_

### Official Review · Reviewer_3R87 · 2021-07-16

**Rating:** 6
**Confidence:** 4

**Summary:**

This paper proposes an algorithm that achieves minimax regret $O(\sqrt{(B_{\star}^2+B_{\star})SAK})$ for SSP with stochastic costs. Their algorithm is based on a novel reduction from general SSP to a loop-free MDP, and a new finite horizon algorithm whose leading term in the regret depends polynomially on the expected cost of the optimal policy and only logarithmically on the horizon.

**Ethical Concerns:**

This work is mainly theoretically and has no ethical concerns.

**Limitations And Societal Impact:**

This work is mainly theoretically and has no foreseeable societal impact.

**Main Review:**

This paper resolves an important question of SSP with stochastic costs: removing the extra $\sqrt{S}$ factor in the leading term.
The proposed loop-free reduction is simple and its analysis is insightful. The matching lower bound is quite interesting from my perspective. The writing is mostly clear and easy to follow.

There are two main shortcomings of this work in my opinion: 1. the obtained result seems to have a worse constant term compared to Tarbouriech et al. [2021] which has logarithmic dependency on the horizon even in the lower order term. Moreover, their method does not require loop-free reduction. Do you think the techniques  of Tarbouriech et al. [2021] can be applied here to achieve horizon-free regret?
2. the finite horizon algorithm seems to be a simple extension of EULER [Zanette and Brunskill, 2019], thus has limited novelty.

Overall, I think the problem studied in this paper is important and the results are complete. Thus, I think it is an acceptable submission.

**Questions**
1. Can the proposed method be directly extended to linear function approximation?
2. Can you elaborate why the loop-free reduction used in Chen and Luo [2021] is not applicable here? It seems to me that if we run Bernstein-SSP at the end of the finite horizon MDP an set $H_2=2B_{\star}$, we are also able to get the optimal regret bound. Moreover, the constant term becomes $O(B_{\star}^{3/2}S^2A)$, which has no dependency on $T_{\star}$?

**Reference**

[Zanette and Brunskill, 2019]: Tighter problem-dependent regret bounds in reinforcement learning without domain knowledge using value function bounds

**Time Spent Reviewing:**

3

---

> ### Author Response · Authors · 2021-08-10
> **Response to Reviewer 3R87**
>
> We thank the reviewer for this thorough review. Regarding the shortcomings:
>
> 1. It remains an interesting open problem whether one can obtain horizon-free guarantees in SSP with a reduction to finite-horizon. On the other hand, it important to remember that the reduction has a clear computational advantage over the method of Tarbouriech et al. [2021] - the total computational complexity of the reduction grows logarithmically with the number of episodes $K$, compared to polynomially in their method. We see both methods as important contributions to the SSP literature.
>
> 2. This paper makes a significant contribution to the finite-horizon literature: we show that properly tuning exploration bonuses can make significant improvement in the regret, making it scale polynomially in the expected cumulative cost of the optimal policy rather than with the horizon H (which can be arbitrarily larger). This is an important contribution since previous algorithms required much stronger assumptions (that the total reward in every trajectory is bounded by 1), while our analysis -- unlike previous algorithms -- does not require any additional assumption.
>
> Regarding the questions:
>
> 1. Recently, the paper “Regret Bounds for Stochastic Shortest Path Problems with Linear Function Approximation” (Vial et al.) showed that it is possible to extend some of the methods from tabular SSP to linear SSP. However, linear function approximation creates many challenges in SSP and requires the authors to come up with new techniques and additional assumptions, e.g., all policies are proper. Extending SSP to linear function approximation is a very interesting question, and we believe that our reduction could play a major role in that thanks to its generality.
>
> 2. This is a good question. Using the reduction of Chen and Luo [2021] results in a regret bound of $\sqrt{T_\star B_\star S A K}$ which is optimal for adversarial costs but not for stochastic costs. That is, if we run Bernstein-SSP in the end of the finite-horizon MDP then the leading term in our regret will scale with $T_\star$ and not $B_\star$ as the result we obtained. Our reduction avoids this dependence by iteratively applying the finite-horizon algorithm until the goal is reached.
> Moreover, note that the lower order term in the finite-horizon regret already has a polynomial dependence in $T_\star$, so this will not be avoided anyway in the reduction. There are two important open questions here regarding the lower order term. First, does it need to be polynomial in $H$ for finite-horizon - we already know that this can be avoided under the strong assumption that the total reward in every trajectory is bounded by 1, but here we do not make any such assumptions. Second, can the SSP to finite-horizon reduction avoid polynomial $T_\star$ dependence in the lower order term - this is left for future work as the focus of this paper is establishing the minimax optimal regret for SSP.

---

### Official Review · Reviewer_tvpW · 2021-07-16

**Rating:** 7
**Confidence:** 3

**Summary:**

As the title suggests, this paper established the minimax regret for the stochastic shortest path (SSP) problem. Specifically, given a state space S, action space A, an upper bound B_\star on the total cost of the shortest path, the rate-optimal regret (up to logarithmic factors) with K episodes is of the order \Theta(sqrt((B_\star^2 + B_\star)SAK)). This result improves over the prior result by a factor of sqrt(S) in the upper bound, as well as a factor of sqrt(B_\star) in the lower bound if B_\star < 1. The main contributions of this paper are:

1. The authors proposed a novel black-box reduction from SSP to finite-horizon MDPs, which holds for a general class of finite-horizon MDP regret minimization algorithms.

2. Similar to the celebrated optimistic algorithms, in finite-horizon MDP the authors proposed the ULCVI algorithm for regret minimization, where the key technical novelty is to upper bound the regret in terms of the optimal value function instead of the horizon.

3. For B_\star < 1 the authors also proved a matching minimax lower bound.

**Main Review:**

I like this paper. This paper completely characterized the minimax regret for the SSP problem, in terms of the optimal dependence on all parameters (S,A,K,B_\star). Although the reduction idea is very intuitive, the analysis is still technical, and the authors extracted widely applicable conditions such that the reduction holds. The ULCVI algorithm for regret minimization in finite-horizon MDPs are could be of interest on its own. The writing is also very clear.

Just some minor comments:

1. Section 3.1 discussed the case of unknown B_\star, then how about unknown T_\star if there is no meaningful c_\min? Remark 2 partially addressed this question, but I'd like to see whether there is a technical idea to tackle it provably or currently we can only do practical tricks.

2. Could you provide a comparison with the additional terms in the SSP literature?

3. Page 3, Line 117: technically speaking B_\star \le D may not hold as B_\star is only an upper bound.

Post-rebuttal feedback:

Thanks to the authors for the response, and I've also read other reviewers' comments. This paper settles the minimax regret for the stochastic shortest path problem, which I think is a solid contribution and recommend it for acceptance.

**Time Spent Reviewing:**

3

---

> ### Author Response · Authors · 2021-08-10
> **Response to Reviewer tvpW**
>
> We thank the reviewer very much for this kind review.
>
> 1. Please see comment to Reviewer f8JW regarding unknown $T_\star$. Here is a technical idea to tackle this issue, but analyzing it is left for future work as the main focus of this paper is establishing the minimax optimal regret for SSP. Note that you can always assume there is some $c_{\min}$ by capping the costs from below (as done in Rosenberg et al. [2020]) therefore obtaining an upper bound on $T_\star$ of $B_\star / c_{min}$. Now, you could run CORRAL (Agarwal et al., ‘16) with $log( B_\star / c_{min})$ copies of our algorithm, each with its own value for $T_\star$ (with exponentially growing values of $T_\star$), and obtain a regret bound against that estimated $T_\star$ up to constant factor. Since the dependence on $c_{min}$ in the regret bound is only logarithmic, you can set $c_{min} = 1 / poly(K)$, and have $T_\star$ appear only in low-order terms.
>
> 2. We assume you refer to the difference between $B_\star$ and the diameter $D$. This difference is discussed in detail in Rosenberg et al. [2020]. Briefly, in SSP, time-dependent quantities such as the diameter do not capture the hardness of the problem as tightly as cost-dependent quantities such as $B_\star$. This is since the optimal policy can run for an extremely long time until reaching the goal, but have a tiny cumulative cost overall.
>
> 3. This is a nice comment. You are correct that $B_\star$ may be larger than $D$ if it is not a tight upper bound, but in our algorithm we can always estimate the diameter (similarly to the way we estimate $B_\star$) and make sure that $B_\star$ is smaller. We will definitely add a discussion about this to the camera-ready version of our paper.

---

### Official Review · Reviewer_WiEG · 2021-07-17

**Rating:** 7
**Confidence:** 4

**Summary:**

The paper proposed a novel reduction from stochastic shortest path (SSP) to finite-horizon Markov decision process (MDP). Based on this approach, the paper proposes a model-based algorithm, showing that the minimax regret is $\tilde{O}(\sqrt{(B_{\star}^2+B_{\star})|S||A|K})$, where where $B_{\star}$ is a bound on the expected cost of the optimal policy from any state, $S$ is the state space, and $A$ is the action space. This upper bound attains the lower bound $\Omega(\sqrt{B_{\star}^2|S||A|K})$ for $B_{\star} \ge 1$. The paper also prove a matching lower bound $\Omega(\sqrt{B_{\star}|S||A|K})$ for $B_{\star} < 1$. In addition, the paper provides a new algorithm for regret minimization in finite-horizon MDPs and the regret depends only logarithmically on the horizon length.

**Main Review:**

Strengths:

1. The paper proposes a novel black-box reduction from SSP to finite-horizon MDPs.

2. The paper improves the minimax regret bound by $|S|$.

3. The paper proves a tighter lower bound for $B_{\star}<1$ and the upper bound matches the lower bound in both cases, $B_{\star}<1$ and $B_{\star}\ge 1$.

4. The regret bound depends only logarithmically on the horizon.

5. The paper considers the case where the costs are i.i.d. and initially unknown instead of deterministic and known cost function.

Weaknesses

1. The proposed reduction approach only supports model-based algorithms, which also need to satisfy some additional conditions.

2. In the statement of Theorem 2.3, the lower bound $\Omega(\sqrt{B_{\star}|S||A|K})$ is for the case $B_{\star}\le 1/2$. What if $1/2 < B_{\star} < 1$?

3. The algorithm needs to have the knowledge of $T_{\star}$, which is the upper bound on the expected time steps of the optimal policy, as mentioned in Remark 2, Page 4.

Other comments:

1. Page 3, Line 124-126: The reviewer thinks that the regret depends logarithmically on the horizon $H$ from the theorem.

2. Page 5, Algorithm 1: Is the terminal cost $\hat{c}_f$ occurred at $h=H$ or $h=H+1$ for each interval? In Algorithm 1, only $c(s_t,a_t)$ are fed to the algorithm $\mathcal{A}$. It is not clear whether $\hat{c}_f$ is included. The reviewer thinks that more details should be included in Algorithm 1.

3. Page 8, Line 304-307: $B_{\star}$ is already defined by the upper bound on the cost of the optimal policy for the SSP as in Line 116, but here is another definition. The reviewer thinks that it is very confusing. Also, in Line 306, it is not clear why we can assume $B_{\star} \le H$.


**Time Spent Reviewing:**

16

---

> ### Author Response · Authors · 2021-08-10
> **Response to Reviewer WiEG**
>
> We thank the reviewer for this kind review. Regarding the weaknesses:
>
> 1. Our black box reduction assumes a model-based algorithm which satisfies certain assumptions. Nevertheless, these assumptions are quite standard and most current state-of-the-art algorithms satisfy these assumptions (e.g., EULER and ORLC).
>
> 2. Note that for $1/2 \le B_\star \le 1$ we have that $B_\star$ and $B_\star^2$ are similar up to a constant factor, so this does not matter in terms of the regret.
>
> 3. As discussed in Remark 2, it remains an important open question whether the reduction needs to know $T_\star$ in advance. In this paper we focused on establishing the exact minimax regret for SSP, and establishing whether additive polynomial dependence in $T_\star$ is necessary is also left for future work. However, as discussed in Remark 3, $T_\star$ is an important parameter in practice because it controls the computational complexity of the algorithm and it is available many times.
>
> Regarding the other comments:
>
> 1. This is correct - the regret depends only logarithmically on the horizon $H$ (for large enough $K$). This will be made clear in the camera-ready version.
>
> 2. The terminal cost occurs in step $h=H+1$. $\hat c_f(s^m_{H+1})$ is not fed to the algorithm $\mathcal{A}$ because it knows the terminal costs $\hat c_f$ and therefore can compute $\hat c_f(s^m_{H+1})$ given the trajectory. To make this clearer, we will feed it to $\mathcal{A}$ as well.
>
> 3. Thank you for pointing that out. We will rename $B_\star$ in Section 5 to avoid confusion. Note that costs are bounded between $[0,1]$ so it is always the case that $B_\star \le H$.

---

### Official Review · Reviewer_f8JW · 2021-07-21

**Rating:** 6
**Confidence:** 3

**Summary:**

The paper focuses on the problem of stochastic shortest path. The authors provide a regret bound of $\tilde{O}(\sqrt{({B_*}^2+B_*)SAK })$ , which matches the lower bound up to log terms. Besides, the authors show a lower bound of $\Omega(\sqrt{B_* SAK})$ when $B_*< 1$. Their results hold even in the case $B_*$ is unknown.

**Ethical Concerns:**

There is no apparent ethical issues.

**Ethics Review Area:**

["I don’t know"]

**Limitations And Societal Impact:**

Yes.

**Main Review:**

Pros
The final regret bound is tight up to logarithmic terms and the algorithm can work without knowing $B_*$.

Cons
The algorithm is mainly based on previous algorithms for finite-horizon RL and most techniques are similar. That is, the novelty is not enough.

Other questions:
1. Have you ever tried the algorithm with only $\underline{J}$? In this way the analysis could be simplified a lot since there is no need to deal with the gap between $\underline{J}$ and $\bar{J}$. In particular, given the knowledge of $B_*$, we can restrict $\underline{J}$ to $[0, B_*]$ to avoid dependence on $H$.

2. Is the parameter $T_*$ necessary in the algorithm? Moreover, can we avoid the dependence on $T_*$ (even in the lower order terms)?

**Time Spent Reviewing:**

4

---

> ### Author Response · Authors · 2021-08-10
> **Response to Reviewer f8JW**
>
> We thank the reviewer for the thorough review. We want to emphasize that this paper presents two important novel techniques. First, the reduction from SSP to finite-horizon MDPs is able to achieve optimal regret. This reduction required us to develop new techniques which significantly depart from the finite-horizon analysis. Second, we also make an important contribution to the finite-horizon literature: we show that properly tuning exploration bonuses can make significant improvement in the regret, making it scale polynomially only with the expected cumulative cost of the optimal policy and not the horizon $H$ (which can be arbitrarily larger). This is an important contribution since previous algorithms required much stronger assumptions (that the total reward in every trajectory is bounded by 1), while our analysis -- unlike previous algorithms -- does not require any additional assumption.
>
> Regarding the questions:
>
> 1. Similarly to other finite-horizon optimistic algorithms (e.g., EULER), maintaining both an upper bound and a lower bound on the optimal value function is crucial in order to obtain tight regret guarantees. An example for this can be seen in the last term of our exploration bonus in Eq. (2). This term is decreasing because the upper and lower estimates become close to each other. However, without also maintaining the upper bound, this term will stay large if for example the optimal value function is much smaller than $B_\star$.
>
> 2. As discussed in Remark 2, it remains an important open question whether the reduction needs to know $T_\star$ in advance. In this paper we focused on establishing the exact minimax regret for SSP, and establishing whether additive polynomial dependence in $T_\star$ is necessary is also left for future work. However, as discussed in Remark 3, $T_\star$ is an important parameter in practice because it controls the computational complexity of the algorithm and it is available many times.

---

### Decision · Program_Chairs · 2021-09-27

**Decision:**

Accept (Poster)

**Comment:**

This paper makes solid contribution to the stochastic shortest path problem.
Reviewers are all in favor of acceptance.
Please do incorporate all the suggestions from the reviews into the final version.